# Structural basis of mRNA decay by the human exosome–ribosome supercomplex

Alexander Kögel[1,4], Achim Keidel[1,4], Matina-Jasemi Loukeri[1], Christopher C. Kuhn[1], Lukas M. Langer[1], Ingmar B. Schäfer[1,2,3✉] & Elena Conti[1✉]

The interplay between translation and mRNA decay is widespread in human cells[1–3]. In quality-control pathways, exonucleolytic degradation of mRNA associated with translating ribosomes is mediated largely by the cytoplasmic exosome[4–9], which includes the exoribonuclease complex EXO10 and the helicase complex SKI238 (refs. 10–16). The helicase can extract mRNA from the ribosome and is expected to transfer it to the exoribonuclease core through a bridging factor, HBS1L3 (also known as SKI7), but the mechanisms of this molecular handover remain unclear[7,17,18]. Here we reveal how human EXO10 is recruited by HBS1L3 (SKI7) to an active ribosome-bound SKI238 complex. We show that rather than a sequential handover, a direct physical coupling mechanism takes place, which culminates in the formation of a cytoplasmic exosome–ribosome supercomplex. Capturing the structure during active decay reveals a continuous path in which an RNA substrate threads from the 80S ribosome through the SKI2 helicase into the exoribonuclease active site of the cytoplasmic exosome complex. The SKI3 subunit of the complex directly binds to HBS1L3 (SKI7) and also engages a surface of the 40S subunit, establishing a recognition platform in collided disomes. Exosome and ribosome thus work together as a single structural and functional unit in co-translational mRNA decay, coordinating their activities in a transient supercomplex.

The ribosome has emerged as a key player in governing both the life and the death of cytoplasmic mRNAs[1–3]. Data across different eukaryotic organisms have shown that beyond orchestrating the factors required for translating mRNAs into proteins, ribosomes can also directly interact with mRNA-degrading factors and engage in co-translational mRNA decay[4–9]. In human cells, a major route of ribosome-associated mRNA decay revolves around quality-control pathways[6]. These pathways activate after detecting faulty translation progression, triggering the degradation of problematic mRNAs and the recycling of ribosomes stalled on them[3]. The ribonuclease in these processes is the RNA exosome complex, which degrades transcripts in a 3′–5′ direction and in a processive manner[10,11]. In cytoplasm, the human ribonuclease core complex (EXO10) contains nine catalytically inert subunits (EXO9) and the active subunit DIS3L (refs. 12,13). Substrate targeting to EXO10 requires the recruitment of a dedicated helicase complex that is formed by the association of the RNA helicase SKI2 and the catalytically inactive subunits SKI3 and SKI8 (ref. 14). Mutations in the genes that encode exosome subunits and cofactors have been linked to a range of diseases[15,16], underscoring their physiological importance.

The human SKI2–SKI3–SKI8 (SKI238) complex directly interacts with 80S ribosomes[7,17], engaging the 40S subunits with a recognition mechanism that is overall conserved from yeast to humans[7,17,18]. By default, SKI238 is mainly in a 'closed' conformation, with the 3′–5′ RNA-unwinding channel of the SKI2 helicase blocked by the so-called gatekeeping module, which is formed by the N-terminal domain of SKI2, the tetratricopeptide repeat (TPR) protein SKI3 and two β-propeller SKI8 subunits[17,19] (referred to as SKI2$_N$38 here). Consistent with the autoinhibited state of the closed conformation, binding of SKI238 to ribosomes generally does not hinder productive translation elongation[7]. In the event of aberrant stalling or termination, however, SKI238 can extract the mRNA 3′ end in an ATP-dependent manner[7]. This process is connected to a conformational switch of SKI238 into an active 'open' state, whereby the SKI2$_N$38 gatekeeping module detaches from the helicase module (SKI2$_H$, comprising the catalytic DExH core and the regulatory arch domain; referred to as SKI2$_{H/DExH}$ in the figures)[17]. At this stage, the RNA 3′ end can exit the helicase and, in principle, enter the exosome ribonuclease cage. However, exosome recruitment requires an additional protein. The necessity of a bridging factor to couple the cytoplasmic exosome ribonuclease and helicase complexes was first shown in budding yeast[20,21], in which the molecular and regulatory mechanisms that underlie this connection have been elucidated[22,23]. Whether a similar choreography of interactions exists in humans remains uncertain, given the different domain structure and limited sequence conservation of the Ski7 functional homologue, which has been identified as a splicing isoform of HBS1L (HBS1L3; referred to as SKI7 here)[22,24]. More generally, it is unclear whether the 15-subunit human cytoplasmic exosome complex can

[1]Department of Structural Cell Biology, Max Planck Institute of Biochemistry, Martinsried, Germany. [2]Present address: Paul Langerhans Institute Dresden and Center of Membrane Biochemistry and Lipid Research, Faculty of Medicine, TU Dresden, Dresden, Germany. [3]Present address: German Center for Diabetes Research, Neuherberg, Germany. [4]These authors contributed equally: Alexander Kögel, Achim Keidel. ✉e-mail: ischaefe@biochem.mpg.de; conti@biochem.mpg.de

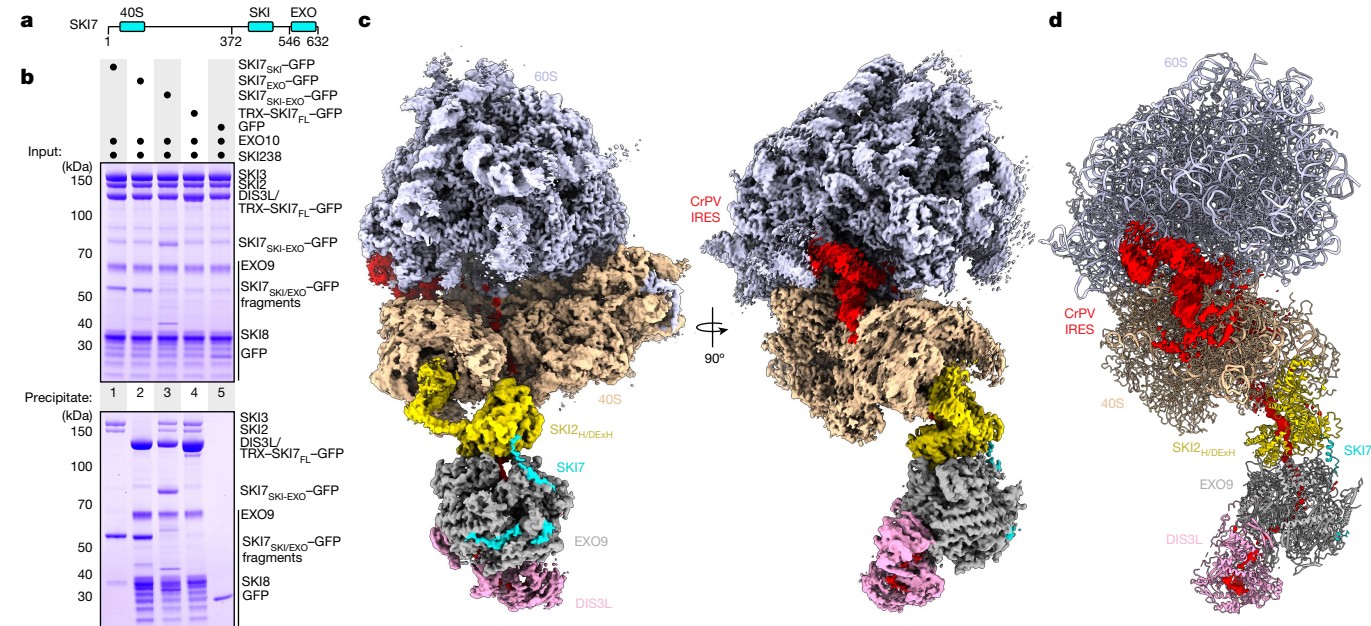

**Fig. 1 | Structure of the human cytoplasmic exosome–ribosome supercomplex. a**, Schematic representation of the organization of the human SKI7 domain, with the binding domains for SKI238 (SKI), EXO10 (EXO) and 40S identified in this study (see also **b** and Extended Data Fig. 1). **b**, Identification of the SKI-EXO-binding region of SKI7 by pull-down assays with purified recombinant SKI238 and EXO10 complexes through GFP-tagged SKI7 (fragments or full-length (FL), as indicated). Mixed recombinant proteins (input, top panel, lanes 1–5) and proteins retained on GFP beads (precipitate, bottom panel, lanes 6–10) were analysed by SDS–PAGE. **c**, Single-particle

cryo-EM composite map showing the 80S ribosome with the 60S coloured in light blue, 40S in sand, SKI2 helicase in yellow, EXO9 subunits in grey, DIS3L in pink, SKI7 in cyan and RNA substrate in red. The overall resolution estimates of the focused reconstruction are 3.2 Å, 3.3 Å and 3.7 Å for the 80S, 40S and cytoplasmic helicase–exosome assemblies, respectively. **d**, Model of the cytoplasmic EXO10-SKI-80S complex shown in cartoon representation, coloured as in **c**. Density for RNA is shown as an omit map displayed in a 30-Å radius around the IRES in red.

assemble in the ribosomal context as an exosome–ribosome super-complex, or whether the helicase dissociates from the ribosome beforehand.

## Human SKI7 integrates exosome activities

Human SKI7 and HBS1L are protein variants that are encoded by the same gene and generated by alternative splicing. The two isoforms share an identical N-terminal domain (1–144 residues), but contain distinctive C-terminal domains that have different functions, binding to either the RNA-degrading exosome[22,24] or the ribosome-splitting factor PELOTA[25] (Extended Data Fig. 1a). To assess the abundance of SKI7 at the protein level, we analysed a previous deep proteome study[26]. Although it is less abundant than HBS1L, the SKI7 isoform is present at levels similar to those of the cytoplasmic exosome subunits DIS3L and SKI3 (Extended Data Fig. 1b). Thus, the abundance of the SKI7 bridging factor is not limiting and is sufficient to sustain the capacity of the cytoplasmic exosome pathway.

Previous biochemical studies[22,24] identified the region responsible for EXO9 binding in the C terminus of SKI7 (SKI7$_{EXO}$) (Fig. 1a and Extended Data Fig. 1a). In binding assays with purified recombinant proteins, we found that SKI238 interacts with the fragment immediately N-terminal to SKI7$_{EXO}$ (residues 372–546; referred to as SKI7$_{SKI}$ here) (Extended Data Fig. 1c). Consistently, GFP-tagged SKI7$_{SKI}$ co-precipitated SKI238 but not EXO10 (Fig. 1b, lane 6) and GFP-tagged SKI7$_{EXO}$ co-precipitated EXO10 but not SKI238 (Fig. 1b, lane 7). Recapitulating these results, a GFP–SKI7$_{SKI-EXO}$ protein that spans both functional regions was capable of co-precipitating both SKI238 and EXO10 (Fig. 1b, lane 8, compare with full-length in lane 9). Thus, the fundamental architectural framework of the cytoplasmic exosome bridging factor, relying on two distinct and independent functional

domains, has been retained from yeast to humans[20,22,23]. This conservation across evolution is notable given that not only the detailed sequence has diverged, but even the overall position of the domains (that is, at the N terminus of yeast Ski7 and at the C terminus of human SKI7; Extended Data Fig. 1a). Of note, similar pull-down assays showed that a shared N-terminal fragment of SKI7 and HBS1L interacts specifically with the ribosome 40S subunit (Extended Data Fig. 1d), which suggests that the two isoforms compete for 40S binding rather than for SKI238 binding.

## Reconstitution of human 80S-bound EXO-SKI

Having identified the SKI7$_{SKI-EXO}$ region sufficient to bridge the interaction between SKI238 and EXO10, we proceeded to reconstitute and visualize the corresponding 15-subunit human cytoplasmic exosome (EXO10–SKI7$_{SKI-EXO}$–SKI238; called EXO10-SKI) in the context of a ribosome-bound substrate (EXO10–SKI7$_{SKI-EXO}$–SKI238–80S; called EXO10-SKI-80S).

To mimic an mRNA substrate engaged with a human 80S ribosome, we used an in-vitro-transcribed RNA with the internal ribosome entry site (IRES) from the cricket-paralysis virus (CrPV) and an additional 29-nt single-stranded sequence at the 3′ end of the P site (CrPV-IRES RNA[17]) (Extended Data Fig. 1e). To assess the suitability of CrPV-IRES as an exosome–ribosome substrate, we added a Broccoli (Brc) RNA aptamer at the 5′ end and monitored the integrity of Brc-CrPV-IRES in RNA-degradation assays by fluorescence. When incubating the Brc-CrPV-IRES-80S sample with an active EXO10-SKI (that is, with wild-type SKI2 and DIS3L), we observed that the fluorescence signal of the Brc-CrPV-IRES RNA faded in the presence of ATP but not in its absence (Extended Data Fig. 1f), suggesting that 80S-bound CrPV-IRES is indeed a suitable substrate for EXO10-SKI.

## Structure of human 80S-bound EXO-SKI

For structural characterization, we incubated the 80S-bound CrPV-IRES RNA with ATP and with EXO10-SKI containing an active SKI2 and an inactive DIS3L (D486N mutant), with the rationale of extracting the RNA substrate from the 80S but preventing degradation when the substrate 3′ end would reach the exosome exoribonuclease site. The sample was reconstituted (Extended Data Fig. 2a and Extended Data Table 1) and subjected to single-particle cryo-electron microscopy (cryo-EM) analysis (Extended Data Fig. 2b–g). Signal subtraction of the 80S combined with focused three-dimensional (3D) classification, subsequent refinement and post-processing resulted in a reconstruction with nominal resolution ranging from 3.3 Å at the core to 8.7 Å in the outer regions.

The composite map showed the $SKI2_H$ sandwiched between the 40S subunit and the $EXO10-SKI7_{EXO}$ complex (Fig. 1c). Density for the RNA substrate was visible for the entirety of the binding path in the supercomplex, and was generally well defined in the ribosome (except for PK-1), in $SKI2_H$ and at the entry into the exosome ribonuclease (Fig. 1d and Extended Data Fig. 2h). The poorly defined density for the PK-1 pseudoknot of CrPV-IRES corroborates the expectation that it would unfold when the 3′ end of this substrate is pulled into the exosome (Fig. 1d and Extended Data Fig. 1e,f). The cryo-EM density was interpreted by fitting existing atomic models (EXO9 (refs. 27,28); 40S, 80S, SKI238, CrPV-IRES PK-2 and PK-3 (ref. 17)) and by modelling aided by predictions with AlphaFold-Multimer[29] (DIS3L, RRP45 C-terminal domain ($RRP45_{CTD}$) and $SKI7_{EXO}$) (Fig. 1d).

## RNA path from 80S to helicase to RNase

Human EXO9 has the typical barrel-like structure, with a lower ring of six RNase PH-like proteins (RRP41, RRP42, RRP43, RRP45, RRP46 and MTR3) and an upper ring of three S1–KH ('cap') proteins (RRP4, RRP40 and CSL4)[10,11] (Fig. 2a). A notable difference from nuclear EXO9-containing complexes is that a long segment of the $RRP45_{CTD}$ (residues 281–351) becomes structured as it wraps around roughly half of the EXO9 outer circumference (Extended Data Fig. 3a). The reorganization of the $RRP45_{CTD}$ seems to be connected to the presence of two specific features of the cytoplasmic complex, DIS3L and SKI7 (Extended Data Fig. 3a), as described below.

The DIS3L subunit contains an endonuclease-like PIN domain and an RNase II-like exoribonuclease domain that interact with the lower ring of EXO9 (Fig. 2a). DIS3L adopts a conformation similar to that observed[27,28] for nuclear DIS3, but presents two unique features as compared with DIS3. First, the DIS3L PIN domain has a compromised active site, as already noted[30], suggesting that it functions exclusively as a structural component in the human cytoplasmic exosome. Second, DIS3L contains a C-terminal domain that binds to EXO9 by inserting between the RNase PH domain and the $RRP45_{CTD}$ (Fig. 2b and Extended Data Fig. 3b). Notwithstanding the distinct structural features on the outer surface of the human cytoplasmic exosome, the RNA-binding path inside the central chamber remains similar to that of its nuclear counterpart[27,28].

The outer surface of EXO10 presented additional densities, which we could attribute to three separate patches of $SKI7_{EXO}$. In the first patch, SKI7 binds with a short helix (amino acids 555–572) in a cleft at the RRP43–MTR3 interface (Fig. 2c and Extended Data Fig. 3c), consistent with mutagenesis data[24], and is further encased by the $RRP45_{CTD}$ (Extended Data Fig. 3c). The second binding patch involves a loop of SKI7 (residues 582–593) that fits between RRP43 and an extended segment of the $RRP45_{CTD}$ (Fig. 2c and Extended Data Fig. 3d). The third binding patch is located at the EXO9 upper ring, where SKI7 (residues 602–616) binds at the cap protein CSL4 (Fig. 2d and Extended Data Fig. 3e). From here, SKI7 residues 617–632 protrude into solvent with a highly conserved α-helix that extends towards the SKI2 catalytic core, engaging it in direct interactions at the RecA1 domain. Overall, $SKI7_{EXO}$

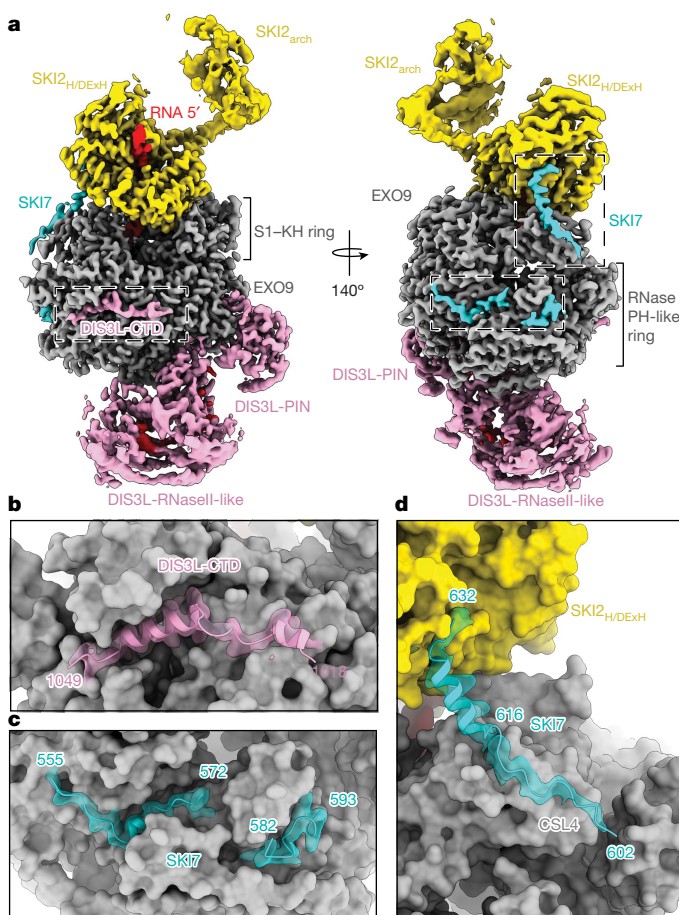

**Fig. 2 | General and distinctive features of the ribosome-bound cytoplasmic exosome. a**, Focused cryo-EM reconstruction of EXO10-SKI bound to the 40S subunit of the cytoplasmic EXO10-SKI-80S complex. Colours are as in Fig. 1c,d. Different domains of DIS3L and the EXO9 barrel are indicated. **b–d**, Magnified views of specific architectural features discussed in the text. The structural models of the $DIS3L_{CTD}$ (**b**) and $SKI7_{EXO}$ (**c**,**d**) are shown in cartoon representation, fitted in the transparent cryo-EM density. The other parts of the complex are shown with the structural model in non-transparent surface representation.

follows a binding path similar to that of the yeast orthologue[22], and also incorporates a SKI-binding element. Notably, the SKI7-binding site on $SKI2_H$ is sterically accessible only in the absence of the $SKI2_N38$ gatekeeping module; that is, in the open-state conformation of the helicase complex.

The catalytic core of SKI2 interacts with the exosome cap proteins in an edge-on position, such that the RNA 3′ end exiting from the helicase channel is directed towards the entry of the EXO9 chamber (Fig. 2a). This spatial arrangement, as well as the RNA-binding path, are overall similar to those in nuclear MTR4 (refs. 27,31) and yeast Ski2 (ref. 23). The opposite surface of the DExH core and the arch domain both interact with the 'head' of the 40S subunit (Fig. 2a), similar to previous structures that did not include the exosome[17,18]. In this spatial configuration, the 3′ end of the substrate exiting from the mRNA-binding channel in the ribosome points towards the entry of the SKI2 helicase channel.

## SKI3 is a docking platform for SKI7

The structure of the human cytoplasmic exosome–ribosome supercomplex described above did not capture two key elements: $SKI7_{SKI}$ (which is required for recruiting SKI238 to $EXO10-SKI7_{EXO}$ in the biochemical assays; Fig. 1b); and the $SKI2_N38$ gatekeeping module. Reasoning that human $SKI7_{SKI}$ might bind to the $SKI2_N38$ gatekeeping module,

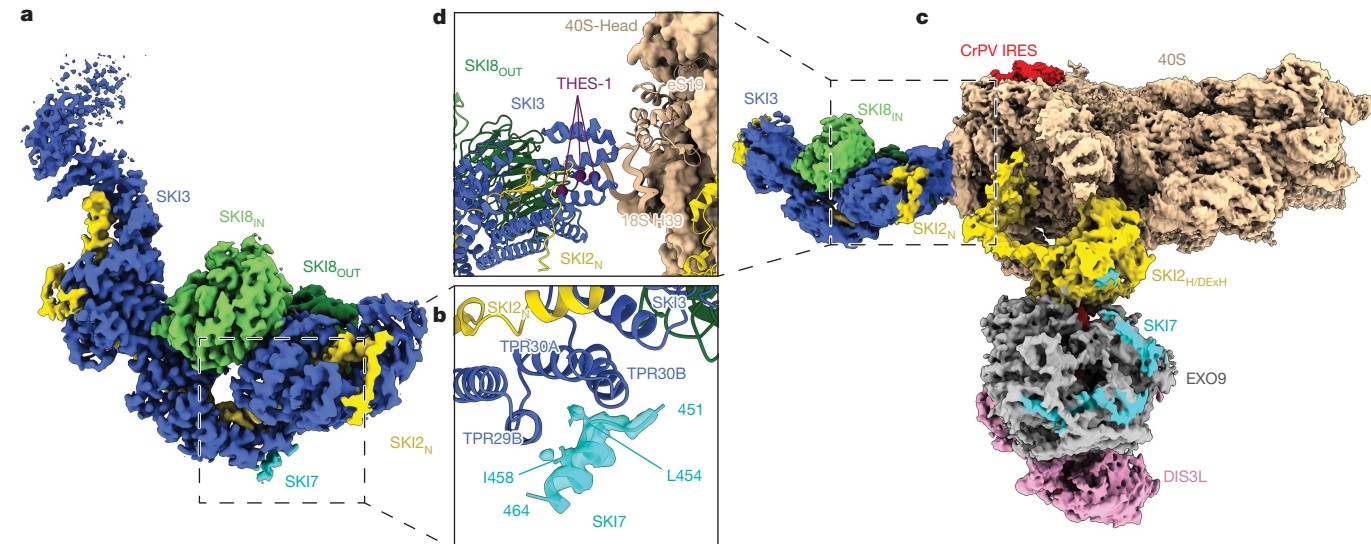

**Fig. 3 | The SKI2$_N$38 gatekeeping module is an interaction platform.**
**a**, Single-particle cryo-EM reconstruction of SKI2$_{\Delta arch}$38 bound to SKI7$_{SKI}$ at a global resolution estimated at 3.4 Å. The SKI238 complex is in an open conformation, with ordered density only for the SKI2$_N$38 gatekeeping module (SKI3 is in blue, SKI2$_N$ in yellow, SKI8$_{IN}$ and SKI8$_{OUT}$ in light and dark green, respectively, and SKI7 in cyan). **b**, Magnified view of the SKI7$_{SKI}$-binding site on SKI3. Density is in transparent representation. Residues L454 and I458 mutated in the assay in Extended Data Fig. 4h are highlighted. The SKI3 TPR helices 29B, 30A and 30B correspond to residues 1105–1115, 1123–1134 and 1139–1150, respectively. **c**, Single-particle cryo-EM composite map coloured as in **a** and in Fig. 1c. The overall resolution estimates of the focused reconstruction are 3.4 Å, 4.1 Å and 6.5 Å for the 40S subunit, the SKI2$_N$38 gatekeeping module and the cytoplasmic helicase–exosome assemblies, respectively. **d**, Magnified view of the interaction site between the SKI2$_N$38 gatekeeping module (in cartoon representation) and the 40S subunit. For clarity, the 40S-interacting regions are shown in cartoon representation and all other 40S components are highlighted. Colours are as in **c**. Residues that are mutated in patients with THES (L1485, R1503 and L1505) are shown as purple spheres.

we proceeded to visualize this interaction in the context of a minimal complex. To favour the open conformation of SKI238, in which the gatekeeping module is dissociated from the helicase module[17], we purified a complex that lacks the SKI2 arch domain (SKI2$_{\Delta arch}$38) and assembled it with SKI7$_{SKI}$ (Extended Data Fig. 4a). Cryo-EM analysis of this sample resulted in the reconstruction of the complex in the open state at a nominal resolution of 3.4 Å (Extended Data Fig. 4b–g), revealing a small unexplained patch of density on the convex surface of the SKI3 superhelix (Fig. 3a). To build an atomic model in this density, we used the knowledge of the protein boundaries we had obtained experimentally to inform targeted predictions with AlphaFold-Multimer[29]. These predictions reproducibly converged on a model in which an SKI7 helix (residues 451–464) fitted seamlessly into the experimental cryo-EM density, with two hydrophobic residues (Leu454 and Ile458) binding in the cleft of SKI3 formed between TPRs 29 and 30 (Fig. 3b). This model was validated by in vitro pull-down experiments with recombinant wild-type and mutant SKI7$_{SKI}$ proteins (Extended Data Fig. 4h, compare lanes 5 and 4, L454D/I458D wild-type and mutant, respectively). It is worth noting that although AlphaFold predicts an additional helix of SKI7 (Extended Data Fig. 5a,d), we found that it made no substantial contribution to binding.

The interacting residues between SKI7 and SKI3 are evolutionarily conserved in higher eukaryotes, but not in budding yeast[23], which diverged in terms of both the general structural elements and the detailed amino acid sequences that underpin the Ski3–Ski7 interaction (Extended Data Fig. 5). Notably, however, lower and higher eukaryotes converged on the same regulatory mechanism; namely, that the recognition of the exosome bridging factor on the outer surface of the TPR scaffold protein is favoured when the helicase complex is in the open-state conformation.

## The SKI2$_N$38 module binds colliding ribosomes

When analysing the cryo-EM data and, in particular, a subset of 40S-containing particles, we recognized the characteristic shape of

the SKI2$_N$38 gatekeeping module in a density feature on the side of the 40S subunit (Extended Data Fig. 6a). Albeit poorly defined, the same feature could also be detected in the context of the 80S-containing particles (Extended Data Fig. 6b). Processing of the 40S-containing subset (Extended Data Fig. 6c) allowed us to obtain a reconstruction showing that the ribosomal subunit binds to the helicase–ribonuclease exosome assembly through the SKI2$_H$ module (as described before for the 80S-bound reconstruction) and to the SKI2$_N$38 gatekeeping module (Fig. 3c and Extended Data Fig. 7). Fitting the structure of SKI2$_N$38 in the density (Fig. 3d and Extended Data Fig. 7g) revealed that the C-terminal repeats of human SKI3 (TPRs 39–40) interact with the 'head' of the 40S subunit, in particular contacting the 18S rRNA (at helix 39 in expansion segment 9) and ribosomal protein eS19 (Fig. 3d). Notably, this region of SKI3 is a hotspot for mutations associated with a congenital disease known as trichohepatoenteric syndrome (THES)[32]. The accumulation of disease-related mutations at SKI3 TPR39 (in the so-called THES-1 hotspot[17]; Fig. 3d) suggests that the 40S-binding region of the SKI2$_N$38 gatekeeping module is physiologically important.

The binding site of the SKI2$_N$38 gatekeeping module on the 40S subunit has some implications in the context of ribosome stalling—a functionally relevant scenario considering that SKI2 has been found to be enriched on transcripts after ribosome stalling events in cells[6]. When we superposed our reconstruction with that of a collided disome[33], we noticed that in a configuration in which the helicase and gatekeeping modules of SKI238 would bind to the stalled (leading) ribosome, the SKI8$_{OUT}$ subunit would inherently be positioned to interact with the trailing ribosome (Fig. 4a and Extended Data Fig. 8). To test the structure-based hypothesis that the open-state conformation of SKI238 recognizes collided disomes, we performed polysome profiling experiments by inducing translation stalling in the rabbit reticulocyte lysate (RRL) in vitro system with the previously described Xbp1-XTEN mRNA[33]. Ribosome nascent chain complexes (RNCs) were incubated with a recombinant SKI2$_{\Delta wedge}$38 mutant complex that was previously shown to favour the open-state conformation[17], or with a recombinant SKI2$_N$38

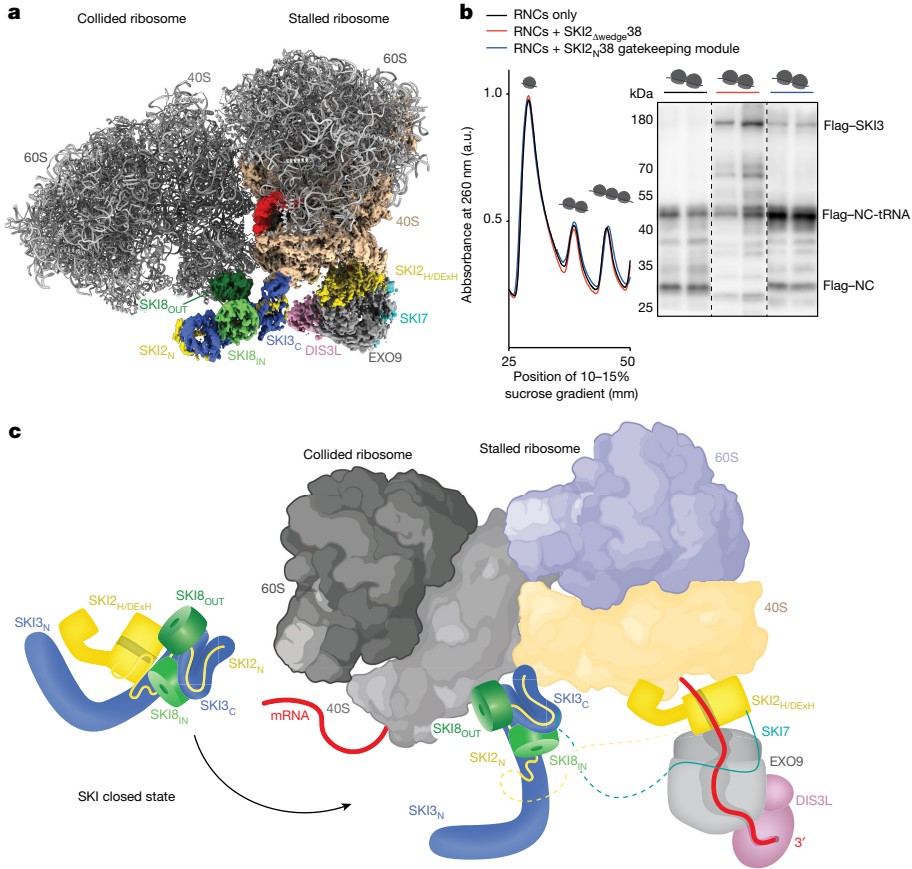

**Fig. 4 | The cytoplasmic exosome–ribosome supercomplex is compatible with disome engagement. a**, Superposition of the EXO10-SKI-40S composite map (shown in Fig. 3c) with the structure of a mammalian disome stalled on an XBP1 reporter mRNA[33]. The disome is shown in cartoon representation and coloured in light grey (Protein Data Bank (PDB): 7QVP)[33]. All other colours are as in Fig. 3c. **b**, Sucrose gradient profiles (10–50% sucrose gradient) (left) and anti-Flag western blots (right) of the disome-containing gradient fractions of Xbp1-XTEN stalled RNCs with the SKI2$_N$38 gatekeeping module (blue), Xbp1-XTEN stalled RNCs with SKI2$_{\Delta wedge}$38 (red) and stalled RNCs only (control; black). a.u., arbitrary units; NC, nascent chain. **c**, Schematic depiction. Left, closed state of the SKI238 complex[17]. Right, structural analysis of the cytoplasmic exosome–ribosome supercomplex with the SKI238 complex in an activated open state. The mRNA substrate (red) is being channelled from the 80S through the SKI2 helicase to the exosome DIS3L ribonuclease for degradation. In the model, the architecture of the exosome–ribosome supercomplex in RNA-degradation mode is compatible with the geometry of a collided ribosome (in grey), with the SKI2$_N$38 gatekeeping module wedging in the composite surface formed by the stalled and collided 40S subunits (also see Extended Data Fig. 8). The schematic was created with BioRender.com.

gatekeeping module that lacks the 80S-SKI2$_H$ interaction, before being subjected to ultracentrifugation on a sucrose density gradient and compared with a control RNC-only sample (Fig. 4b and Extended Data Fig. 9a). The Flag tag encoded in the reporter (approximately 45-kDa peptidyl-tRNA product) and in recombinant SKI3 (around 175 kDa) allowed us to then analyse the polysome profile fractions by anti-Flag western blotting (Fig. 4b and Extended Data Fig. 9b). These biochemical analyses show that both the open-state SKI2$_{\Delta wedge}$38 complex and the SKI2$_N$38 gatekeeping module can be enriched in the disome fractions from sucrose gradients of stalled RNCs (Fig. 4b and Extended Data Fig. 9b), suggesting a direct interaction.

## Conclusions and perspective

In this work, we visualize the structure of a human cytoplasmic exosome–ribosome supercomplex, and provide mechanistic insights into the coupling between mRNA degradation and translation-dependent quality-control pathways. The structural information reveals the intricate and conserved regulatory mechanisms that underlie the formation of an active cytoplasmic exosome, capturing the complex as it extracts a ribosome-bound substrate by means of the SKI2 helicase and transfers it to the DIS3L exoribonuclease. In the active 'open' state

of the helicase complex, the regulatory SKI2$_N$38 gatekeeping module not only recruits the cytoplasmic exosome by binding to SKI7, but also engages the ribosome 40S subunit. The structural and biochemical analyses furthermore suggest that the SKI2$_N$38 gatekeeping module can bind collided disomes. These results lead to a model in which colliding ribosomes offer multiple surfaces to collectively facilitate the targeted recruitment and activation of the SKI238 helicase complex on a stalled ribosome, establishing the conditions for channelling the 3′ end of the relevant mRNA to the degradation activity of the exosome (Fig. 4c). Notably, the same disome surfaces are also used for ubiquitylation during ribosome-associated quality control[33,34] (Extended Data Fig. 8c). The presence of mutually exclusive interactions at colliding ribosomes might thus regulate the interplay between mRNA degradation and nascent polypeptide degradation.

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

# Methods

## Cloning, protein expression and purification

The complete open reading frames of SKI2 (SKIV2L, UniProt: Q6PGP7), SKI3 (TTC37, UniProt: Q15477) and SKI8 (WDR61, UniProt: Q9GZS3) were obtained from a human cDNA library (MegaMan Human Transcriptome Library, Agilent Technologies) and cloned in separate expression cassettes on a single plasmid (pACEBac1)[35]. SKI3 was designed with a cleavable 10×His tag at the N terminus. The complex was expressed in baculovirus-infected Hi5 insect cells and purified by nickel affinity and ion-exchange chromatography. All derivatives, that is, SKI2$_N$38, SKI2$_{\Delta arch}$38 and SKI2$_{\Delta wedge}$38, were expressed and purified similarly. A detailed protocol can be found in a previous study[17].

The complete open reading frame of SKI7 (HBS1L isoform 3, UniProt: Q9Y450; also described as SKI7$_{FL}$ here) was commercially synthesized (Eurofins Genomics) and cloned with N-terminal 6×His-TRX-3C and C-terminal GSG-eGFP-TwinStrep tags under an IPTG-inducible promotor for expression in *Escherichia coli* (pEC-A-6xHis-TrxA-3C-SKI 7-eGFP-TwinStrepII construct). Transformed *E. coli* BL21 STAR pRARE cells were grown in TB medium at 37 °C under antibiotic selection to an optical density at 600 nm (OD$_{600}$) of 2. The temperature was reduced to 18 °C and protein expression was induced by the addition of 500 µM IPTG. Cells were collected after 16 h by centrifugation at 8,500$g$ and resuspended in lysis buffer containing 20 mM HEPES pH 7.5, 300 mM NaCl, 25 mM imidazole and 2 mM β-mercaptoethanol (β-ME), and supplemented with 200 U ml$^{-1}$ benzonase (Merck), 500 µM AEBSF protease inhibitor and cOmplete EDTA-free protease inhibitor cocktail (Roche). The cells were lysed by ultrasonication (Bandelin, Sonopuls basic). The recombinant protein was kept strictly at 4 °C and the purification procedure was performed quickly to avoid degradation. The lysate was cleared by centrifugation and loaded on a HisTrap HP 5-ml column (Cytiva) for nickel affinity chromatography. The column was washed with 15 column volumes of 20 mM HEPES pH 7.5, 1,000 mM NaCl, 200 mM KCl, 10 mM MgCl$_2$, 25 mM imidazole and 2 mM β-ME, followed by washing with 5 column volumes of lysis buffer. The protein was eluted with lysis buffer supplemented with 350 mM imidazole into a StrepTrap HP 5-ml column (Cytiva) for a second affinity step. The bound protein was washed with 10 column volumes of buffer containing 20 mM HEPES pH 7.5, 300 mM NaCl and 2 mM DTT, and the addition of 5 mM desthiobiotin (DTB) eluted the protein. The protein was concentrated in 10% glycerol (v/v), flash frozen in liquid nitrogen (LN$_2$) and stored at −80 °C.

The SKI7$_{SKI-EXO}$ construct (residues 369–632) was cloned with a N-terminal 6×His-TRX-3C tag and also with a C-terminal 3C-eGFP-TwinStrep tag. The construct behaved similarly to SKI7$_{FL}$ and was quickly purified with the same protocol to prevent degradation.

The shorter SKI7 constructs (residues 1–144 (here also SKI7$_{40S}$), 136–277, 267–386, 372–546 (here also SKI7$_{SKI}$) and 540–632 (here also SKI7$_{EXO}$)) were cloned similarly with N-terminal 6×His-TRX-3C and C-terminal GSG-eGFP-TwinStrep tags. The constructs expressed well and had no tendency for degradation. An initial affinity chromatography step followed by cleavage of the N-terminal 6×His-TRX tag during overnight dialysis combined with a second nickel affinity step to remove the cleaved 6×His-TRX tag resulted in sufficiently pure constructs as judged by Coomassie-stained SDS–PAGE. The N-terminal 6×His-TRX tag of SKI7$_{EXO}$ (residues 540–632) was not cleaved, because its presence helped to avoid protein precipitation during the final concentration step. For cryo-EM sample reconstitution, TRX–SKI7$_{SKI}$ was also purified without cleavage of the TRX tag.

The complete open reading frame of DIS3L (UniProt: Q8T64F) was cloned with a cleavable TwinStrep tag at the N terminus into the piggyBac vector and expressed in a HEK 293T stable cell line adapted to grow in suspension. In brief, 400–800 ml of stably transfected cells (10$^6$ cells per ml) were induced with 1 µg ml$^{-1}$ doxycycline for 48 h and collected by centrifugation at 800$g$. Cells were resuspended in lysis buffer containing 20 mM HEPES pH 7.5, 300 mM NaCl and 2 mM DTT, and lysed using a dounce homogenizer. After clearing the lysate by centrifugation, it was loaded on a 5-ml StrepTrap XT column (Cytiva) and washed with high salt (20 mM HEPES pH 7.5, 1,000 mM NaCl, 200 mM KCl, 10 mM MgCl$_2$ and 2 mM DTT) and lysis buffer. The protein was eluted in buffer containing 20 mM Tris-HCl pH 8.0, 150 mM NaCl, 50 mM biotin and 2 mM DTT. A detailed description, including generation of the HEK 293T stable cell line, can be found in a previous study[17].

## Reconstitution of cytoplasmic exosome

Human cytoplasmic exosome (EXO10) was reconstituted from equimolar amounts of pre-assembled EXO9 (purified as previously described[22,28] and freshly purified DIS3L or DIS3L(D486N). After incubation for 30 min on ice, the mixture was concentrated by centrifugation at 3,000$g$ using an Amicon Ultra MWCO100 centricon (Millipore) and purified by size-exclusion chromatography (Superdex 200 Increase 10/300 GL, Cytiva) in buffer containing 20 mM HEPES pH 7.5, 150 mM NaCl and 2 mM DTT.

## Purification of ribosomal subunits

Human ribosomal 40S and 60S subunits were obtained from HEK 293T cells (using adapted protocols[36,37]). In brief, ribosomes were pelleted by ultracentrifugation from cleared HEK 293T cell lysate and resuspended in buffer containing 2 mM puromycin and 500 mM KCl to release the nascent peptide chain and dissociate the ribosomal subunits. The 40S and 60S ribosomal subunits were subsequently separated by sucrose gradient centrifugation and the corresponding 40S and 60S fractions concentrated in buffer containing 20 mM HEPES pH 7.5, 50 mM KCl, 4 mM MgCl$_2$ and 2 mM DTT. A detailed description can be found in a previous study[17].

## Co-precipitation assays

A total of 100 pmol GFP-tagged SKI7$_{FL}$ or derivates (SKI7$_{40S}$ 1–144, SKI7 136–277, SKI7 267–386, SKI7$_{SKI}$ 372–546 and SKI7$_{EXO}$ 540–632) were mixed with 130 pmol SKI238, SKI2$_N$38 or EXO10 in buffer containing 20 mM HEPES pH 7.5, 150 mM NaCl, 0.05% v/v NP40 and 2 mM DTT. After incubation with GFP-binder resin (ChromoTek GFP-Trap Agarose) at 4 °C for 45 min and three washes in the same buffer, the resin was dried and taken up in SDS buffer. Input and precipitate fractions were analysed by Coomassie-stained SDS–PAGE on a NuPAGE 4–12% Bis-Tris gel (Thermo Fisher Scientific). Co-precipitation of 40S and 60S ribosomal subunits was done accordingly, but in buffer containing 20 mM HEPES pH 7.5, 50 mM KCl, 4 mM MgCl$_2$, 0.05% v/v NP40 and 2 mM DTT.

## Intensity-based absolute quantification analysis of proteomic data

Intensity-based absolute quantification (iBAQ) values were collected across five cell lines (GM12878, hESC, HeLaS3, HepG2 and K562) using five different proteases (AspN, GluC, LysC, LysN and trypsin) from a previous proteome study[26]. The data were plotted as box plots in Extended Data Fig. 1b. Here, quartiles are represented by boxes with whiskers extending to the highest or lowest value within 1.5 times the interquartile range. Median values are shown as thick horizontal bars inside the boxes. Values falling further than 1.5 times the interquartile distance are considered outliers and are shown as dots. iBAQ values are plotted in log scale.

## In vitro transcription of RNA substrates

The genomic sequence of the CrPV-IRES was taken from the intergenic region of the CrPV genome (NCBI GeneBank: 6025–6232 nt, NC_003924.1) and fused to a portion of the firefly luciferase open reading frame (modified from a previous report[38]). The used CrPV-IRES construct was amplified by PCR, gel-purified and transcribed in vitro with T7 RNA polymerase by run-off transcription. The transcription reaction was performed in 40 mM Tris-HCl pH 8.0, 28 mM MgCl$_2$, 0.01%

Triton X-100, 1 mM spermidine and 5 mM DTT in the presence of 25 mM of each ribonucleotide (ATP, CTP, GTP and UTP; Jena Bioscience) and 100 U μl⁻¹ T7 RNA Polymerase (MPI Biochemistry Core Facility) at 37 °C for 4 h. The DNA template was digested with 1 U μl⁻¹ DNase I (Roche) and the CrPV-IRES purified by LiCl₂ precipitation. A detailed description can be found in a previous study[17].

The transcription template for the Brc-CrPV-IRES used in the exosome degradation assays was amplified by PCR, fusing the sequence of the broccoli light-up aptamer (GCGGAGACGGTCGGGTCCAGAT ATTCGTATCTGTCGAGTAGAGTGTGGGCTCCGC)[39] to the 5′ end of the CrPV-IRES. Otherwise, the Brc-CrPV-IRES was transcribed as described above.

## Exosome RNA-degradation assays
The 80S-Brc-CrPV-IRES substrate used in the assays was reconstituted by mixing 100 pmol purified human 40S with 120 pmol Brc-CrPV-IRES. After incubation at 37 °C for 5 min, 120 pmol purified human 60S was added, and the incubation was allowed to proceed for another 10 min. The 80S-Brc-CrPV-IRES complex was subsequently purified by sucrose gradient centrifugation in a SW 40 Ti rotor (Beckman Coulter) at 4 °C. Sucrose gradients (15–40%) (w/v) in buffer containing 20 mM HEPES pH 7.5, 50 mM KCl, 4 mM MgCl₂ and 2 mM DTT were mixed using the Gradient Master (Biocomp). After 17 h centrifugation at 22,800 rpm, the gradients were fractionated using a Piston Gradient Fractionator (Biocomp). Fractions corresponding to the 80S-Brc-CrPV-IRES complex were concentrated in buffer containing 20 mM HEPES pH 7.5, 50 mM KCl, 4 mM MgCl₂ and 2 mM DTT by centrifugation at 3,000g using an Amicon Ultra MWCO100 centricon (Millipore). The concentrated 80S-Brc-CrPV-IRES was estimated to be 0.24 μM and used immediately.

Exosome degradation assays were performed in technical triplicates, over a time course of 45 min at 37 °C in buffer containing 20 mM HEPES pH 7.5, 50 mM KCl, 4 mM MgCl₂ and 2 mM DTT, supplemented with 0.5 mM ADP or ATP as indicated in the condition. A single reaction contained 2 pmol 80S-Brc-CrPV-IRES substrate, 2 pmol SKI238, 4 pmol SKI7 and 4 pmol EXO10 in a total volume of 30 μl. Reactions without the addition of EXO10 were included as negative controls for every condition. Time-point samples were taken at 0 min, 5 min, 15 min and 45 min, and the reaction was stopped by adding 10× excess buffer containing 100 mM Tris-HCl pH 7.5, 150 mM NaCl, 300 mM NaOAc pH.5.2, 10 mM EDTA and 1% SDS. The reactions were purified by phenol extraction and ethanol precipitation and separated on a denaturing 12% polyacrylamide gel containing 7 M urea. After incubation of the gel in water to remove urea, the Brc-CrPV-IRES was refolded for 20 min under agitation in the presence of 3,5-difluoro-4-hydroxybenzylidene imidazolinone (Sigma-Aldrich) in buffer containing 40 mM HEPES pH 7.5, 125 mM KCl and 10 mM MgCl₂, and subsequently visualized by fluorescence at a 501-nm wavelength (Typhoon FLA7000, Cytiva). Gel images were quantified by densitometry in ImageJ and plotted with R.

## Cryo-EM sample preparation
The exosome–ribosome supercomplex sample (EXO10-SKI-80S) was prepared by sequentially mixing the individual components, similar to the ribosome-bound samples that we previously described[17]. Sucrose gradient centrifugation and concentration steps resulted in reduced amounts of 80S-bound exosomes according to SDS–PAGE and cryo-EM analysis, and were therefore omitted from the sample preparation procedure. In brief, 100 pmol of purified human 40S was mixed with 100 pmol CrPV-IRES carrying a 29-nt 3′ overhang (referred to throughout the text simply as CrPV-IRES) in buffer containing 20 mM HEPES pH 7.5, 50 mM KCl, 4 mM MgCl₂ and 2 mM DTT. After incubation at 37 °C for 5 min, 110 pmol human 60S was added and incubation proceeded for another 10 min. Next, 100 pmol of purified wild-type SKI238 was added to the 80S-CrPV-IRES complex and incubated for 5 min at 37 °C before adjusting the reaction to 1 mM ATP. After incubation at 37 °C for 15 min, 100 pmol of preincubated SKI7$_{SKI-EXO}$–EXO10(D486N) (equimolar

amounts of EXO10(D486N) and SKI7$_{SKI-EXO}$ incubated for 15 min at 37 °C) was added to the reaction. After 5 min at 37 °C the reaction was adjusted to 2 mM ADP-BeF to stop the ATPase reaction. The incubation was allowed to proceed for another 15 min before adjusting the final volume to 130 μl with the buffer described above and storing the sample at 4 °C. The concentrations of SKI238, SKI7$_{SKI-EXO}$ and EXO10(D486N) were between 12 and 20 μM, leading to only minor alterations in volume and salt concentration in the reaction, and were not compensated for. Aggregated protein was removed by centrifugation at 10,000g and the sample supplemented with 0.04% (v/v) n-octyl-β-D-glucoside before grid preparation. Two times 4-5 μl of the sample were incubated for 1 min on holey carbon grids with carbon support (R2/1, 200 mesh, Quantifoil), that were previously glow-discharged with negative polarity at 20 mA for 30 s using an EMS GloQube (MiTeGen). The sample was plunge-frozen in a liquid ethane–propane mix using a Vitrobot Mark IV (Thermo Fisher Scientific) operated at 4 °C and 95% humidity.

The SKI7-SKI238 sample was prepared by mixing 600 pmol SKI2$_{\Delta arch}$38 with approximately 900 pmol of TRX–SKI7$_{SKI}$ and incubating at 37 °C for 20 min in the presence of 3C protease to cleave the TRX tag. The SKI7$_{SKI}$ fragment (residues 372–546) was difficult to quantify because it does not contain amino acids that would absorb at 280 nm. Therefore, the TRX tag was kept and only cleaved while reconstituting the complex. The sample was then concentrated in buffer containing 20 mM HEPES pH 7.5, 100 mM NaCl and 2 mM DTT to approximately 30 μl. After centrifugation at 18,000g for 10 min to pellet larger aggregates, the sample was injected onto a Superose 6 Increase 3.2/300 column (Cytiva) for size-exclusion chromatography by an Aekta micro (Cytiva). A single peak fraction, containing approximately 500 nM, was collected, and mixed with 0.04% (v/v) n-octyl-β-D-glucoside. Four to five microlitres of the sample was applied to holey carbon grids (R2/1, 200 mesh, Quantifoil), that had been glow-discharged with negative polarity at 20 mA for 30 s using an EMS GloQube (MiTeGen) beforehand. The sample was plunge-frozen in a liquid ethane–propane mix using a Vitrobot Mark IV (Thermo Fisher Scientific) operated at 4 °C and 95% humidity.

## Cryo-EM data collection and processing
The cryo-EM data of the exosome–ribosome supercomplex were collected on a FEI Titan Krios GIII microscope (Thermo Fisher Scientific) at 300 kV, equipped with a Gatan K3 direct electron detector operating in electron-counting mode. The microscope is equipped with a post column energy filter (Gatan GIF Quantum LS) set to a slit width of 10 eV. Images were collected by underfocused acquisition (target range of −0.6 μm and −2.4 μm) at a nominal magnification of 105,000× set up in SerialEM[40] using a beam-tilt-based multi-shot acquisition scheme for faster imaging. This resulted in 48,004 micrograph movies (30 movie frames each) acquired at a pixel size of 0.85 Å per pixel with a total exposure of 64.14 e per Å² spread over 3 s.

On-the-fly micrograph movie processing was assisted by Focus[41], which ran MotionCor2 (ref. 42), GCTF[43] and Gautomatch (https://www2. mrc-lmb.cam.ac.uk/download/gautomatch-053/) on individual images while the data were being collected. Subsequent particle processing was performed in RELION v.3.1 (ref. 44). Eightfold downscaled particles were extracted from the aligned, exposure-weighted micrographs and classified in 2D and 3D to discard non-ribosomal and low-resolution particles from the data. These data were separated on the basis of the results of the 2D classification into one particle stack displaying features of the EXO10-SKI-80S assembly and a second one showing density for the EXO10-SKI-40S assembly only.

The cleaned EXO10-SKI-80S particles (1,189,927) were extracted at the original pixel size of 0.85 Å per pixel and aligned in 3D auto-refinement with a spherical mask using a 40-Å downfiltered starting model based on PDB 4UG0 (ref. 45). The refined particles were then used to subtract a large portion of the 80S ribosome signal from the corresponding particle images. To improve the alignment precision and the quality of the reconstructions in the subsequent steps, the particle images

were recentred on the remaining EXO10-SKI signal and re-extracted in a smaller box in RELION v.3.1 (ref. 44). Subtracted particle images that did not show SKI, EXO10 or leftover ribosome signal were removed in 3D classification and resulted in a set of 565,489 cleaner particles for downstream processing. Further classification in 3D showed a small range of different particles which aligned in classes with $SKI2_H$ complexed by a prominent EXO10 density, with SKI238 without visible exosome density, and in classes that only showed leftover ribosome signal without visible EXO10-SKI density.

In a final step, all EXO10-SKI-containing particles (229,580) were 3D-classified into 6 classes, which resulted in 79,353 well-aligning particles. The ribosome-signal-subtracted particles were aligned by 3D auto-refinement and resulted in a reconstruction of EXO10-SKI at an overall resolution of 3.7 Å (according to the Fourier shell correlation (FSC) cut-off criterion of 0.143)[46]. This map is referred to as the 'EXO10-SKI-80S (80S subtracted)' map, or map1. For masking and map sharpening in the RELION v.3.1 post-processing procedure[44], an ad hoc $b$-factor of −30 was applied. The automatically determined $b$-factor of −110.8 resulted in an oversharpened map with a loss of connectivity particularly in the area of DIS3L. Next, the signal subtraction was reverted and the corresponding EXO10-SKI-80S particles refined to 3.2-Å global resolution (according to the FSC cut-off criterion of 0.143)[46]. An ad hoc $b$-factor of −20 was preferred over the automatically estimated $b$-factor of −71.7 in the RELION post-processing routine[44] to better visualize the 40S ribosomal subunit. This reconstruction is referred to as the 'EXO10-SKI-80S (full map)' map, or map3. The quality of the map in areas of the CrPV-IRES and EXO10-SKI, however was not satisfactory. Therefore, we subtracted the signal of the 60S ribosomal subunit from the reconstruction and aligned the resulting EXO10-SKI-40S particles by 3D auto-refinement, which yielded a reconstruction at 3.3-Å global resolution (according to the FSC cut-off criterion of 0.143)[46]. Masking and map sharpening in RELION post-processing[44] was performed using an ad hoc $b$-factor of −20 (automatically determined $b$-factor −73.6). Although the map quality for EXO10-SKI in this reconstruction improved only marginally, the resolution and volume connectivity for the CrPV-IRES in the intersubunit space improved substantially. This EXO10-SKI-40S reconstruction is referred to as the 'EXO10-SKI-80S (60S subtracted)' map, or map2.

For display and visualization purposes, a composite map of the EXO10-SKI-80S assembly was calculated using map1, map2 and map3 and is shown in Fig. 1. The EXO10-SKI map detailed in Fig. 2 and Extended Data Fig. 3 was produced using DeepEMhancer[47] with the wide target setting.

As described above, the ribosome-bound EXO10-SKI supercomplex data contained a fraction of particles that lacked the entire 60S ribosomal subunit (described as EXO10-SKI-40S here). Two- and three-dimensional classification was performed in RELION v.3.1 (ref. 44) and resulted in 119,144 clean 40S ribosomal particles, which were aligned in 3D auto-refinement for subsequent subtraction of the majority of the 40S ribosome signal. The subtracted particle images were 3D-classified into four classes using a wide mask and a 30-Å low-pass-filtered starting model of the $SKI2_N38$ gatekeeping module (PDB 7QDS)[17]. The classification showed one class with 53,460 particles that shows well-aligning density for the $SKI2_N38$ gatekeeping module. Three-dimensional auto-refinement of the subset gave a reconstruction at 4.1-Å global resolution (according to the FSC cut-off criterion of 0.143)[46]. An ad hoc $b$-factor of −30 was used in the RELION v.3.1 post-processing procedure (automatically determined $b$-factor −134.4)[44]. This reconstruction is referred to as 'EXO10-SKI-40S ($EXO10-SKI2_H$ + 40S subtracted)', or map5. For the corresponding ribosome-bound reconstruction, signal reversion procedures similar to the EXO10-SKI-80S particles above were applied, which led to a EXO10-SKI-40S reconstruction at 3.4-Å global resolution using a manually determined $b$-factor of −20 (automatically determined $b$-factor −68.9). This reconstruction is labelled 'EXO10-SKI-40S

(full map)', or map6. Although the reconstruction shows clear density for the $SKI2_N38$ and $SKI2_H$ helicase modules interacting at two different positions of the 40S subunit, the signal for EXO10 was less coherent. To improve the EXO10 density, repeated signal subtraction of the 40S ribosomal subunit combined with 3D classification on the exosome and signal reversion resulted in 40S ribosome signal-subtracted and signal-reverted 3D reconstructions at 6.5 Å (ad hoc $b$-factor −30, automatically determined $b$-factor −128.3) and 4.7 Å (ad hoc $b$-factor −20, automatically determined $b$-factor −92.5) global resolution, respectively. The reconstruction focused on EXO10-SKI is labelled 'EXO10-SKI-40S (40S + $SKI2_N38$ gatekeeping module subtracted)', or map7, and the signal-reverted reconstruction is named 'Control EXO10-SKI-40S', or map8. The main purpose of the latter was to clarify that in the particle stack showing clear presence for the EXO10-SKI assembly, the $SKI2_N38$ gatekeeping module is also bound to the 40S subunit (see also Extended Data Fig. 7g).

For display and visualization purposes, a composite map of the EXO10-SKI-40S assembly was calculated using map5, map6 and map7 and is shown in Fig. 3.

Data processing of the ribosome-bound human EXO10-SKI supercomplex dataset required signal subtraction of the ribosome to yield reconstructions of comparable interpretability and quality to those described in the manuscript. Classical focused classification and refinement procedures did not yield results of comparable quality.

The EXO10-SKI-80S particles show additional density at the location where the $SKI2_N38$ gatekeeping module binds to the 40S ribosomal subunits in the particles that lack 60S (see Extended Data Fig. 6b). However, the signal is rather weak in comparison, owing presumably to the presence of the 60S. To improve the additional unaccounted density at the 40S head (Extended Data Fig. 6b, left) focused 3D classification using a spherical mask located around the area of interest was performed. The resulting particle stack was used for 3D refinement, resulting in a 3D reconstruction of the 80S ribosome. A low-pass-filtered map of this assembly was used to place the model of $SKI2_N38$ by rigid-body fitting (Extended Data Fig. 6b, right).

The cryo-EM data of the human SKI7-SKI238 complex were collected similarly to the ribosome-bound dataset described above. A total of 20,224 micrograph movies were acquired at a pixel size of 0.85 Å per pixel with a total exposure of 67.8 e per Å[2] spread over 6 s and 40 movie frames. Beam-induced motion correction, CTF estimation, particle picking and processing were done similarly to what was done for the ribosome-bound data above. Fourfold downscaled particles were first extracted from the aligned, exposure-weighted micrographs, and particles that seemed to be non-SKI238 were discarded by 2D and 3D classification. The resulting subset of 422,913 SKI238 particles was extracted at the original pixel size of 0.85 Å per pixel. Further 3D classification into six classes, using a starting model based on a previous cryo-EM structure of the SKI238 complex (PDB 7QDS)[17], gave a final subset of 151,860 particles. Initial 3D auto-refinement gave a 3D reconstruction at an estimated overall resolution of 3.6 Å. The quality of the reconstruction and the level of resolved detail were further improved by Bayesian polishing to correct for per-particle motion and by iterative refinement of the per-particle CTF (taking into account the beam-tilted data-acquisition scheme). This resulted in a final reconstruction with an overall global resolution of 3.4 Å according to the gold-standard FSC (0.143) criterion[46]. Masking and map sharpening using an automatically determined global b-factor of −99.4 were performed in the RELION post-processing routine[44]. This map is referred to here as 'SKI7-SKI238', or map4.

### Density interpretation, model building and refinement

The 80S-bound EXO10-SKI or map1 reconstruction was interpreted by rigid-body fitting pre-existing models of SKI2 (PDB 7QE0)[17], the nuclear human exosome (PDB 6D6Q)[27] and AlphaFold multimer predictions of either $SKI7_{EXO}$ or DIS3L with different human exosome subcomplexes[48].

Initial rigid-body fitting was performed in UCSF Chimera[49]. In several areas of the map (the EXO9 barrel and SKI2$_H$), the resolution and quality of the reconstruction allowed us to manually adjust the fit of the models in Coot[50], followed by real-space refinement from within the PHENIX suite[51]. In peripheral areas of the reconstruction (the DIS3L exonuclease), the resolution and quality of the map was not as good, and required us to use a focused refined map with improved map quality and connectivity, which allowed for rigid-body docking of the AlphaFold model. The electron density for the RNA path inside SKI2$_H$, between SKI2$_H$ and EXO9, and within DIS3L was sufficient to model individual nucleotides. However, the precise sequence of the RNA inside the barrel could not be determined from the reconstruction. Inside the EXO9 barrel and between EXO9 and DIS3L, a previous model (PDB 6D6Q)[27] was used to inform structure building. The final model of the 80S-bound EXO10-SKI complex was validated using MolProbity[52].

The complete model of 80S-bound EXO10-SKI was subsequently assembled by rigid-body fitting of the individual 80S, CrPV-IRES RNA and the above-described EXO10-SKI model into the full reconstruction of the EXO10-SKI-80S (map3) in PHENIX. As a starting model for the human 80S-CrPV-IRES, we used PDB 4UG0 (ref. 45) and PDB 4V92 (ref. 36) as already described in detail previously[17]. After placing the models into the map, the CrPV-IRES was manually adjusted and refined in PHENIX to reflect the lack of PK-1, the connection between the RNA exit of the ribosome and entry into SKI2$_H$ and the channelling through the EXO10-SKI complex.

The model for the reconstruction of the 40S-bound SKI2$_N$38 gatekeeping module from which the signal of the 40S subunit and EXO10-SKI had been subtracted (map5) was interpreted with a pre-existing model of the SKI2$_N$38 gatekeeping module (PDB 7QDS)[17]. Initial rigid-body fitting was performed in UCSF Chimera[49]. Inspection of the fit in Coot[50] showed insufficient density for several TPRs towards the N terminus of SKI3, which were consequently not modelled. Otherwise, the model is in good agreement with the experimental density. The final model was real-space-refined within the PHENIX suite[51] and subsequently validated using MolProbity[52].

The model for the reconstruction of the 40S-bound EXO10-SKI from which the signal of the 40S and the SKI2$_N$38 gatekeeping module had been subtracted (map7) was interpreted by rigid-body fitting of the above-described 80S-bound EXO10-SKI model, real-space-refined within the PHENIX suite[51] and validated using MolProbity[52].

The complete model of the 40S-bound EXO10-SKI was subsequently assembled by rigid-body fitting of the individual models of the 40S subunit, the adjusted model of the CrPV-IRES similar to that described above, the model of the SKI2$_N$38 gatekeeping module and the EXO10-SKI model into the full reconstruction of the 40S-EXO10-SKI (map6) in PHENIX. As a starting model for the human 40S, we used the model of the 40S ribosomal subunit from PDB 4UG0 (ref. 45).

The reconstruction of SKI7-SKI238 (map4) was interpreted with pre-existing models of the SKI2$_N$38 gatekeeping module (PDB 7QDS)[17] and an AlphaFold multimer prediction of a part of the SKI2$_N$38 gatekeeping module in complex with SKI7$_{SKI}$. Initial rigid-body fitting was done in UCSF Chimera[49]. The resolution and quality of the map allowed us to manually modify and refine the models in Coot[50] and subject them to subsequent rounds of real-space refinement from within the PHENIX suite[51]. The final model of human SKI7-SKI238 was validated using MolProbity[52].

## Isolation of stalled RNCs and polysome profiling

The 5′ capped XBP1-XTEN mRNA was obtained by an in vitro transcription reaction from the corresponding DNA template using the mMES-SAGE mMACHINE T7 transcription kit (Thermo Fisher Scientific) as described previously[33]. In vitro translation reactions were performed using the rabbit reticulocyte lysate (RRL) system (nuclease treated, Promega). Translation of the 5′ capped XBP1-XTEN mRNA was performed at 32 °C for 25 min using 0.05 μM XBP1-XTEN template and 0.8 U μl$^{-1}$

RNasin Plus ribonuclease inhibitor (Promega) in a total reaction volume of 1 ml. The stalled RNCs on the 5′ capped XBP1-XTEN mRNA were affinity-purified through the 8×His tag on the nascent polypeptide chain and magnetic beads, as described[33]. The amounts of isolated ribosomes were estimated by UV absorbance at 260 nm. Sucrose density gradients (10–50% sucrose in 50 mM HEPES pH 7.5, 100 mM KOAc, 5 mM Mg(OAc)$_2$, 2 mM DTT, 0.1% octaethylene glycol monododecyl ether (TCI) and 20 U ml$^{-1}$ recombinant RNasin ribonuclease inhibitor (Promega)) were prepared using a Gradient Master (Biocomp). The purified RNCs were incubated with 1.2 × molar excess of SKI2$_N$38 or SKI2$_{\Delta wedge}$38 for 40 min at 4 °C, subsequently applied on a sucrose gradient, and the ribosomal fractions were separated by centrifugation for 3 h at 172,000$g$ and 4 °C in an SW 40 Ti rotor (Beckman Coulter). Polysome profiles were generated by continuous absorbance measurement using an UV absorbance reader (Gilson) and simultaneously fractionated (around 3.39 mm per fraction) with a Piston Gradient Fractionator (Biocomp). For display purposes, the sucrose gradient profiles were superimposed using the highest A260 maximum of the disome peak as a reference.

## Western blotting

Fractions obtained by gradient fractionation were precipitated with trichloroacetic acid and pellets were resuspended in SDS loading buffer. All samples were analysed on 15% SDS–PAGE followed by western blotting onto Immobilon-P$^{SQ}$ transfer membrane (Merck Millipore). Transfer efficiency was monitored by stain-free imaging using the 2,2,2-trichloroethanol in-gel technique. Western blots were probed against monoclonal anti-Flag antibody (Sigma-Aldrich, F3165) in a dilution of 1:5,000 and polyclonal anti-mouse HRP-coupled antibody (Bio-Rad, 172-1011) in a dilution of 1:10,000 in 5% milk, made from 5% w/v milk powder (Roth) in 1× Dulbecco's PBS buffer (DPBS, Thermo Fisher Scientific) and 0.001% Tween20 (Bio-Rad). The chemiluminescence signal was detected using Amersham ECL prime western blotting detection reagent (Cytiva) on a ImageQuant LAS 4000 (GE Healthcare; exposed for 140 s at standard sensitivity).

## Statistics and reproducibility

All in vitro assays (protein co-precipitation (Fig. 1b and Extended Data Figs. 1c,d and 4h); protein purifications and complex reconstitutions (Extended Data Figs. 2a and 4a)), in part or in whole, were successfully reproduced at least twice. The nuclease assays (Extended Data Fig. 1f) were performed as triplicates. For each cryo-EM sample (Extended Data Figs. 2b and 4b), multiple grids were screened before final data collection. The final datasets presented in this article were collected once. The sucrose gradient centrifugations and corresponding western blots (Fig. 4b and Extended Data Fig. 9b) were reproduced twice.

## Reporting summary

Further information on research design is available in the Nature Portfolio Reporting Summary linked to this article.

## Data availability

Cryo-EM density maps that support the findings in this study have been deposited in the Electron Microscopy Data Bank (EMDB) and the PDB under the following accession numbers: map1 (EXO10-SKI-80S (80S subtracted)): EMDB EMD-51133; PDB 9G8N; map2 (EXO10-SKI-80S (60S subtracted)): EMDB EMD-51139; map3 (EXO10-SKI-80S (full map)): EMDB EMD-51132; PDB 9G8M; map4 (SKI7-SKI238): EMDB EMD-51137; PDB 9G8R; map5 (EXO10-SKI-40S (EXO10-SKI2$_H$ + 40S subtracted)): EMDB EMD-51136; PDB 9G8Q; map6 (EXO10-SKI-40S (full map)): EMDB EMD-51134; PDB 9G8O; map7 (EXO10-SKI-40S (40S + SKI gatekeeping module subtracted): EMDB EMD-51135; PDB 9G8P; map8 (control EXO10-SKI-40S): EMDB EMD-51135. All other data are available within the main text or the Extended Data. Raw data and source images are

available in the Supplementary Information. Source data are provided with this paper.

## Code availability

No new code was generated in this study.

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

**Acknowledgements** We thank D. Nedialkova for suggesting the analysis in Extended Data Fig. 1b; J. Cox, P. Sinytcin and F. Bonneau for retrieving and plotting the corresponding data from the deep proteome database; R. Beckmann and T. Inada for the Xbp1-XTEN construct; D. Bollschweiler and T. Schäfer for support with cryo-EM; J. Rajan Prabu and C. Benda for the computational infrastructure; M. Baumgartner, P. Birle, T. Krywcun and D. Wartini for technical assistance; C. Long for help preparing the manuscript; and all members of our groups for suggestions and discussions throughout the project. E.C. acknowledges funding from Max Planck Gesellschaft, the European Research Council (Advanced Investigator Grant EXORICO (740329) and GOVERNA (101054447)), the German Research Foundation SFB1035 and the Novo Nordisk Foundation ExoAdapt Grant (31199). M.-J.L. is funded by a Boehringer Ingelheim Fonds PhD fellowship.

**Author contributions** A. Kögel, A. Keidel and E.C. conceptualized the project. A. Kögel planned and performed experiments. A. Kögel, A. Keidel and I.B.S. processed the electron-microscopy data. M.-J.L. and C.C.K. performed the disome experiments using the RRL system that L.M.L. established in the laboratory. A. Kögel, A. Keidel, I.B.S. and E.C. interpreted the data and wrote the manuscript.

**Funding** Open access funding provided by Max Planck Society.

**Competing interests** The authors declare no competing interests.

**Additional information**
**Correspondence and requests for materials** should be addressed to Ingmar B. Schäfer or Elena Conti.

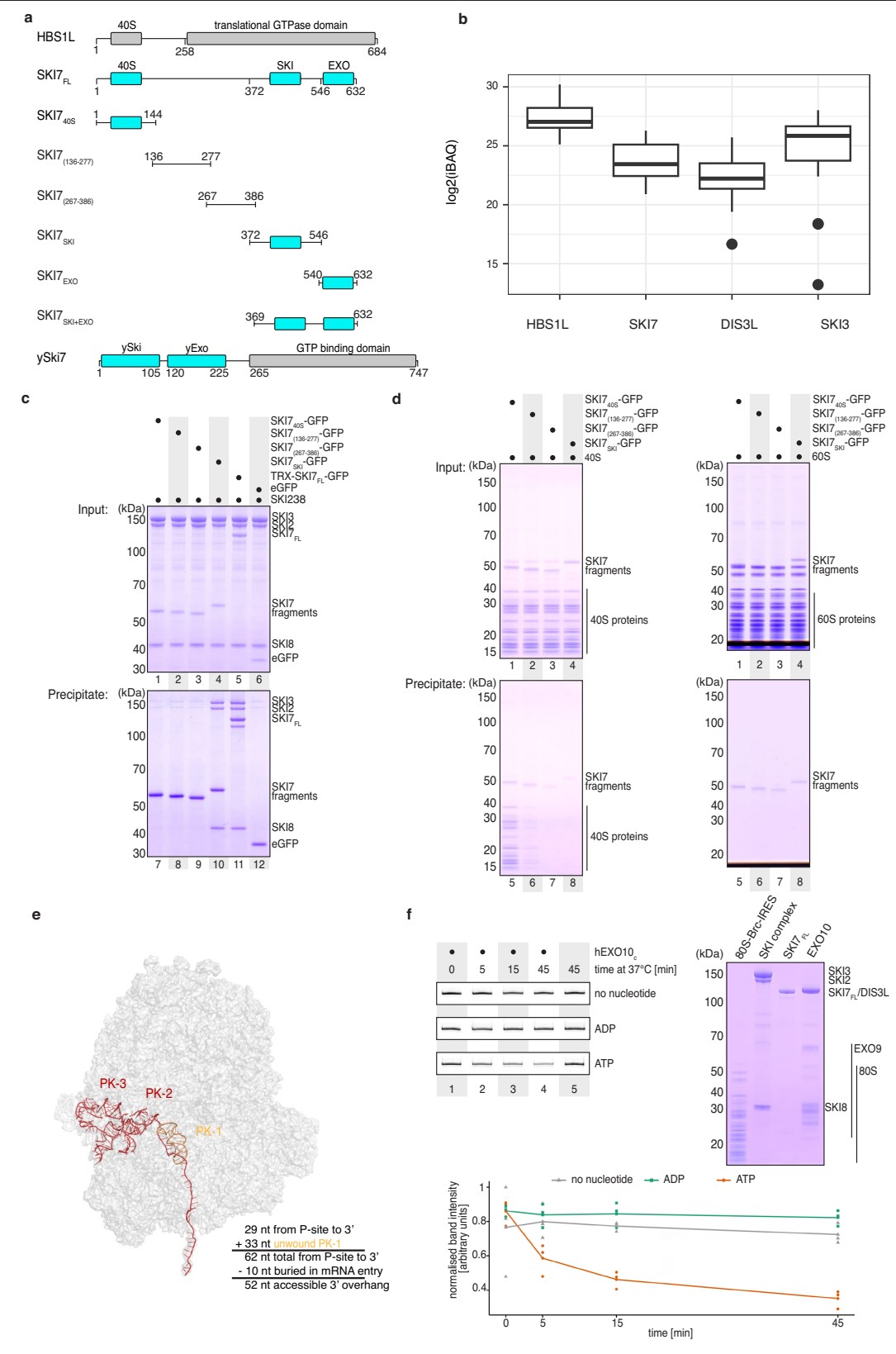

**Extended Data Fig. 1** | See next page for caption.

**Extended Data Fig. 1 | Biochemical characterization of the human cytoplasmic exosome–ribosome supercomplex. a**, Domain organization of human HBS1L and SKI7 with used constructs labelled according to the binding of 40S, SKI and EXO described here. **b**, Distribution of iBAQ[53] values for indicated protein isoforms. N = 25 data points collated from 5 cell lines each treated separately by 5 proteases. Quartiles are represented by boxes with whiskers extending to the maxima or minima value within 1.5 times the interquartile range. Median values are shown as thick horizontal bars inside the boxes. Values falling further than 1.5 times the interquartile distance are considered outliers and are shown as dots. iBAQ values are plotted in log scale. **c**,**d**, GFP protein co-precipitation assays. GFP-tagged SKI7 fragments were mixed with SKI238 complex (**c**) and 40S or 60S subunits (**d**, left and right panels respectively). The Coomassie-stained 4–12% SDS-PAGE gel shows the input on the top and the pulled down protein precipitates at the bottom. **e**, RNA substrate design to trap an active cytoplasmic exosome–ribosome supercomplex. Ribosome structure (grey) bound to CrPV-IRES (red/orange) is from an inactive SKI238-80S[17]. The RNA length from the P site to the ribosomal surface is ~10 nucleotides and that spanning EXO10-SKI ~ 50-55 nucleotides[7,17]. CrPV-IRES will be long enough to reach the DIS3L exoribonuclease if the 3′ pseudoknot is unwound (PK-1, ~33 nucleotides, orange). **f**, RNA-degradation experiments with CrPV-IRES to trap a cytoplasmic ribosome-exosome supercomplex. The left-hand panel shows the RNA-degradation time courses in a 12% UREA-PAGE imaged for the 5′ fluorogenic Broccoli aptamer. The middle panel shows their densitometric quantitation (technical replicates n = 3). Per time point all three individual measurement points are shown. Trend lines connect the respective median. Coomassie-stained 4–12% SDS-PAGE gel with the purified input proteins is on the right.

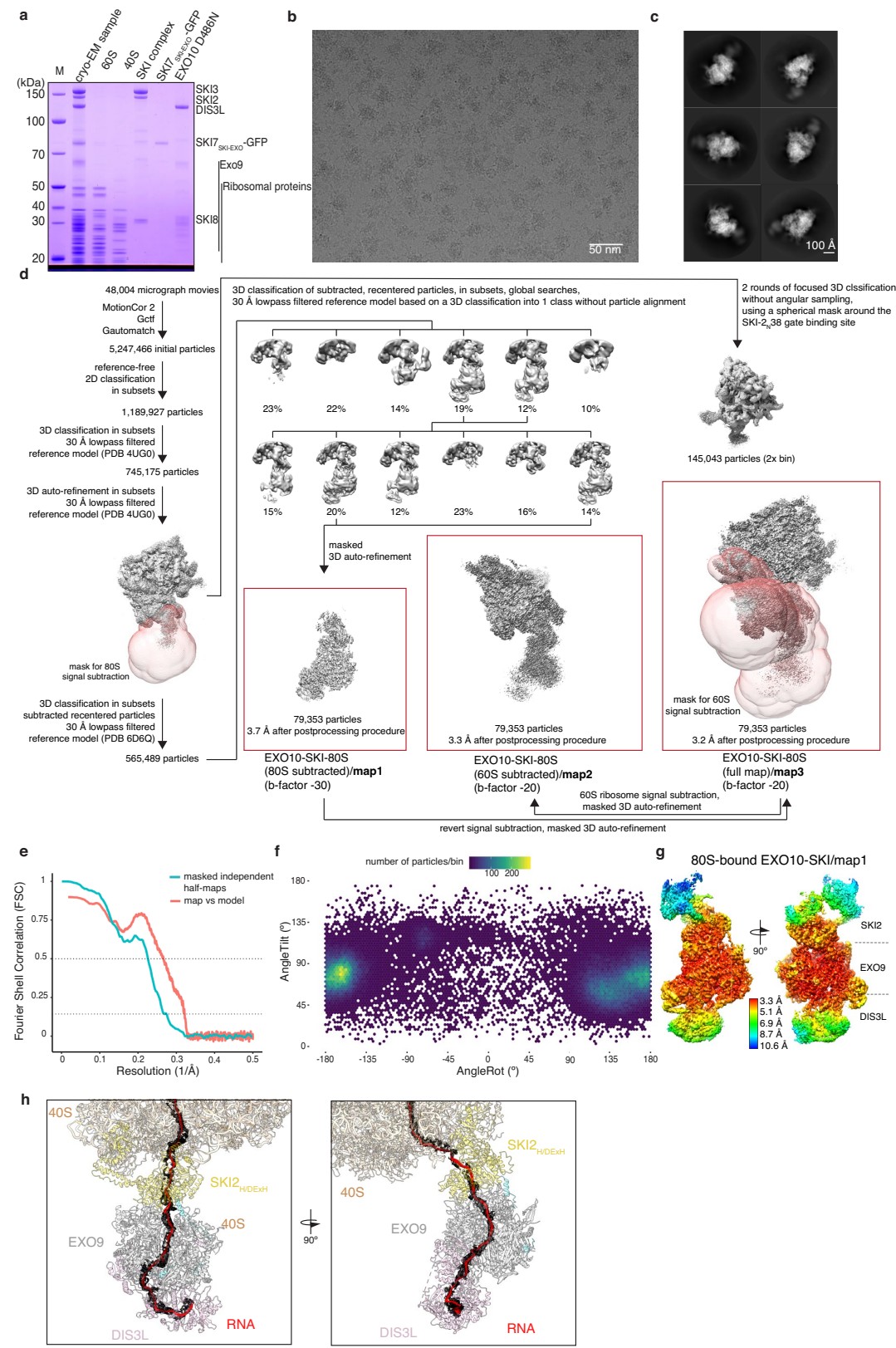

**Extended Data Fig. 2** | See next page for caption.

**Extended Data Fig. 2 | Cryo-EM data analysis of the human cytoplasmic exosome–ribosome supercomplex. a**, A Coomassie-stained 4–12% SDS–PAGE of the complexes used for the reconstitution of the cryo-EM sample. M, molecular weight marker. **b**, A representative cryo-EM micrograph of the human cytoplasmic exosome–ribosome supercomplex sample in A is shown. This image was recorded at 300 kV with a pixel size of 0.85 Å/pixel using a post-GIF K3 direct detector. **c**, 2D class averages of the exosome–ribosome supercomplex. **d**, Processing scheme of the single-particle cryo-EM dataset of the exosome–ribosome supercomplex sample resulting in focused 3D reconstructions for the EXO10-SKI-80S (80S subtracted)/map1 and the EXO10-SKI-80S (60S subtracted)/map2 as well as the full EXO10-SKI-80S (full map)/map3. These three maps were used to calculate the composite map shown in Fig. 1b. The EXO10-SKI-80S map low-pass filtered at 20 Å is shown in Extended Data Fig. 6b. Masks used for the subtraction of partial particle signal are shown in red. **e**, The FSC of the masked and unmasked independent half maps for the EXO10-SKI/map1 reconstruction were calculated in the RELION 3.1 post-processing routine and the map vs model FSC using phenix.mtriage. The FSC cut-off criteria of 0.5 and 0.143 are indicated by dotted lines. **f**, Angular sampling distributions of the EXO10-SKI/map1 reconstruction. Sampling angle data were plotted in 3° by 3° bins and sampling bins coloured according to particle number with yellow indicating more particles and blue fewer particles. **g**, Local resolution estimate of the EXO10-SKI/map1 reconstruction. Position of SKI2, EXO9 and DIS3L are indicated on the right. **h**, Close up on the RNA path within the EXO10-SKI-80S assembly. Model orientation and colours are analogous to Fig. 1c. The model is shown in transparent cartoon representation except for the RNA. The density around the RNA (≤3 Å distance) is shown in black mesh.

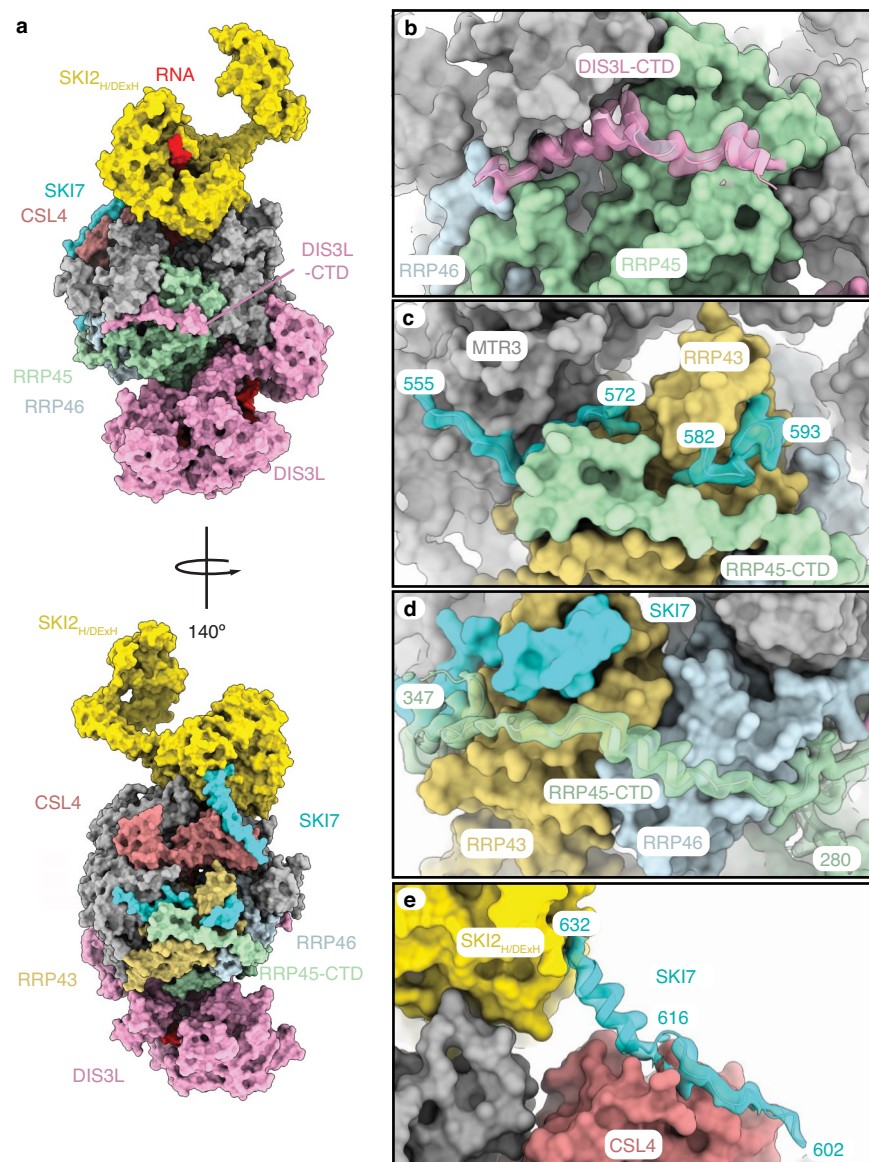

**Extended Data Fig. 3 | Distinctive structural features of the human cytoplasmic exosome. a**, Cryo-EM reconstruction focused on human EXO10-SKI/map1. Colours are as in Fig. 2a except the colouring of specific EXO9 subunits discussed in the text: RRP45 in light green, RRP46 in light blue, RRP45 in light orange and CSL4 in salmon. **b**–**e**, Magnified views of specific architectural features discussed in the text including DIS3L$_{CTD}$ (**b**) SKI7 residues 555 to 593 (**c**), RRP45$_{CTD}$ (**d**) and SKI7 residues 602 to 632 (**e**). The structural models of RRP45$_{CTD}$, DIS3L$_{CTD}$ and SKI7$_{EXO}$ are fitted in the cryo-EM density, shown in a transparent representation. The other parts of the complex are shown with the structural model in surface representation (non-transparent).

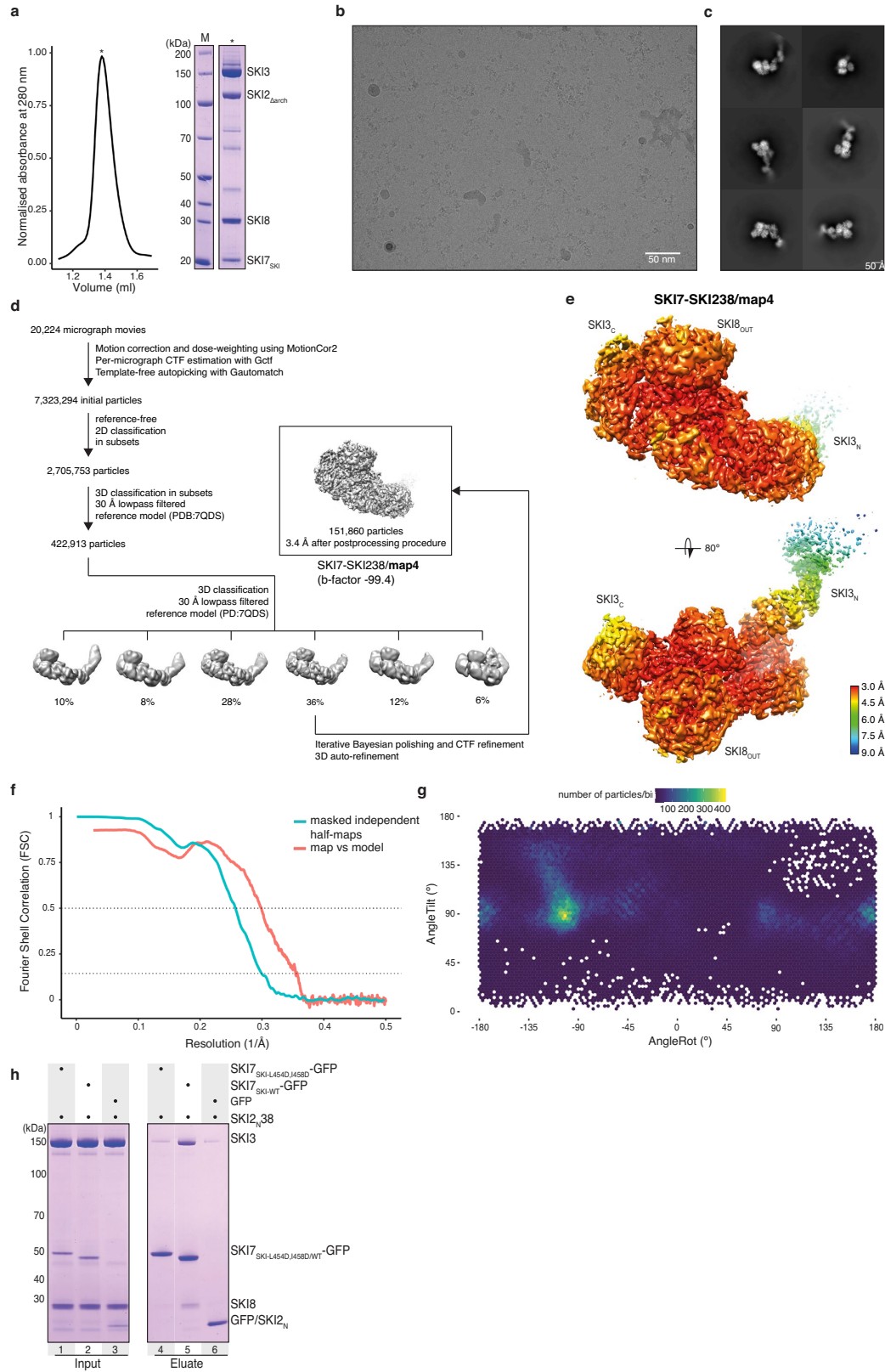

**Extended Data Fig. 4** | See next page for caption.

**Extended Data Fig. 4 | Cryo-EM data analysis of the human SKI7–SKI238 complex. a**, Peak fraction of a Superose 6 gel filtration run (labelled by *; chromatogram on left) of the human SKI7-SKI2$_{\Delta arch}$38 complex analysed on a Coomassie-stained 4–12% SDS–PAGE and later used to prepare SPA (single particle analysis) cryo-EM specimen. M, molecular weight marker. **b**, A representative cryo-EM micrograph of the human SKI7-SKI2$_{\Delta arch}$38 sample in A. This image was recorded at 300 kV with a pixel size of 0.85 Å/pixel using a post-GIF K3 direct detector. **c**, 2D class averages of the human SKI7-SKI2$_{\Delta arch}$38 complex. **d**, Processing scheme of the single-particle cryo-EM dataset of the human SKI7-SKI2$_{\Delta arch}$38 sample resulting in a 3D reconstruction of the complex at an overall estimated resolution of 3.4 Å ("SKI7-SKI238"/map4). This map is shown in detail in Fig. 3a. **e**, Local resolution estimate of the SKI7-SKI238 reconstruction. Red coloration indicates areas where the local resolution is estimated to be highest. **f**, The FSC of the masked and unmasked independent half maps for the SKI7-SKI238 reconstruction were calculated in the RELION 3.1 post-processing routine and the map vs model FSC using phenix.mtriage. The FSC cut-off criteria of 0.5 and 0.143 are indicated by dotted lines. **g**, Angular sampling distributions of the SKI7-SKI238 reconstruction. The data were plotted in 3° by 3° bins and these sampling bins coloured according to particle number with yellow indicating more particles and blue fewer particles. **h**, Pull-down assays with purified recombinant SKI2$_N$38 gatekeeping module via GFP-tagged SKI7$_{SKI}$, either wild-type or with structure-based mutations.

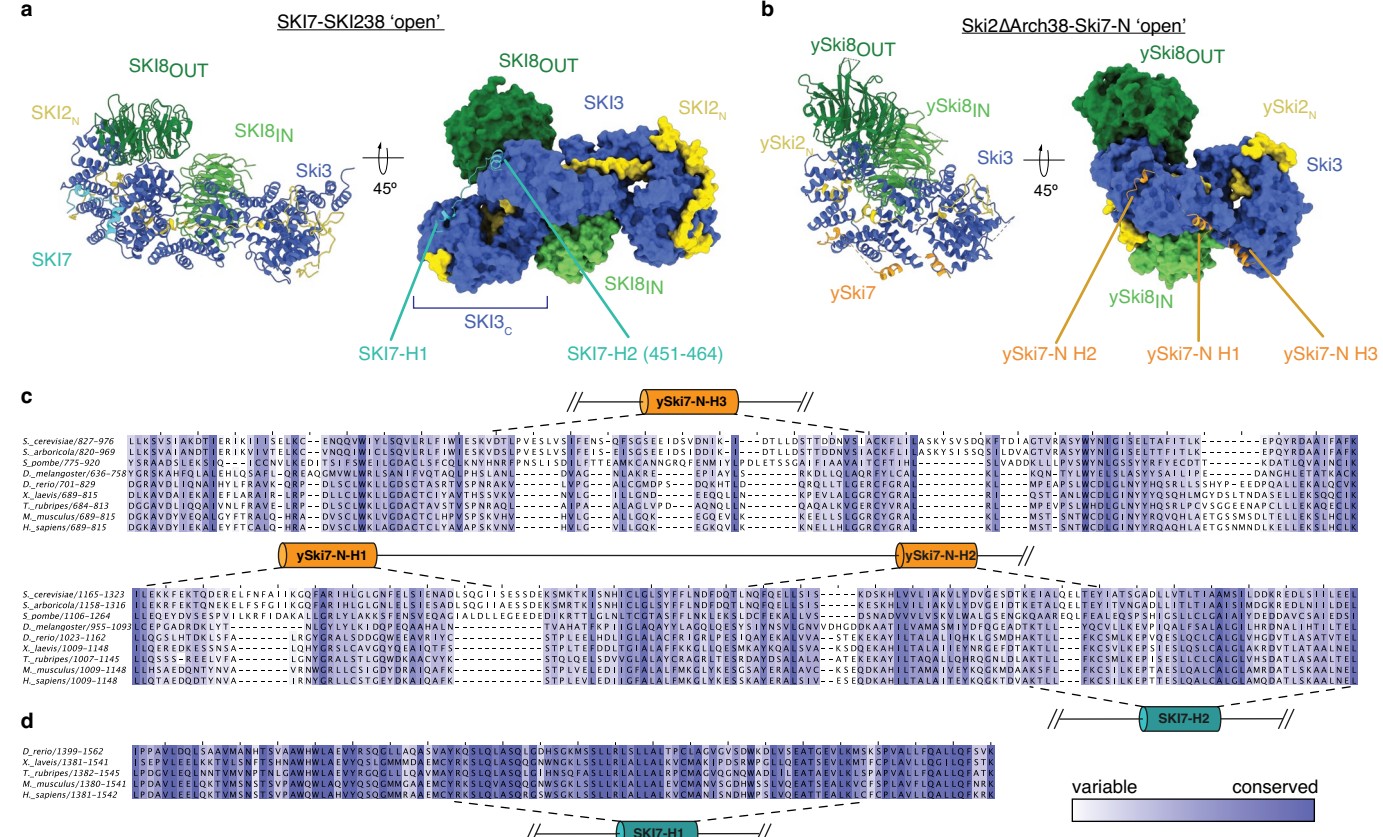

**Extended Data Fig. 5 | Comparison of yeast and human SKI7. a**, Alphafold2 prediction of the human SKI238 complex bound to SKI7 (residues 430–465). Model is shown in two different orientations related by a 90° rotation around a vertical axis. Colours are analogous to Fig. 3a. The model is shown in cartoon representation (left) and surface representation (right). The two helices of SKI7 are labelled SKI7-H1 and SKI7-H2. Helix H2 corresponds to the well-defined helix in the map, is displayed in Fig. 3b and is the major binding determinant. **b**, Model of the ySki238 complex bound to ySki7 (PDB:8QCB)[23]. Colours of the individual subunits and orientation of the model are analogous to **a**. ySki7 is coloured in orange and the three helices are labelled ySki7-H1, ySki7-H2 and

ySki7-H3. **c**, Multiple sequence alignments of SKI3 from different organisms show a lack of conservation at SKI7/ySki7 binding sites. (Organisms with Uniprot identifiers used for alignments: *S. cerevisiae*: P17883, *S. arboricola*: J8PXS5, *S. pombe*: O94474, *D. melanogaster*: Q6NNB2, *D. rerio*: A0A8M1RLS7, *X. laevis*: Q6DFB8, *T. rubripes*: H2RXP3, *M. musculus*: F8VPK0, *H. sapiens*: Q6PGP7). Conservation of residues is indicated by colour from white (variable) to purple (conserved). Binding sites of SKI7 and ySki7 helices are indicated above the alignments with cylinders. **d**, Multiple sequence alignments of the conserved C-terminal domain of SKI3 present in metazoans. (Organisms and Uniprot identifiers analogous to **c** excluding yeast species and *D. melanogaster*.).

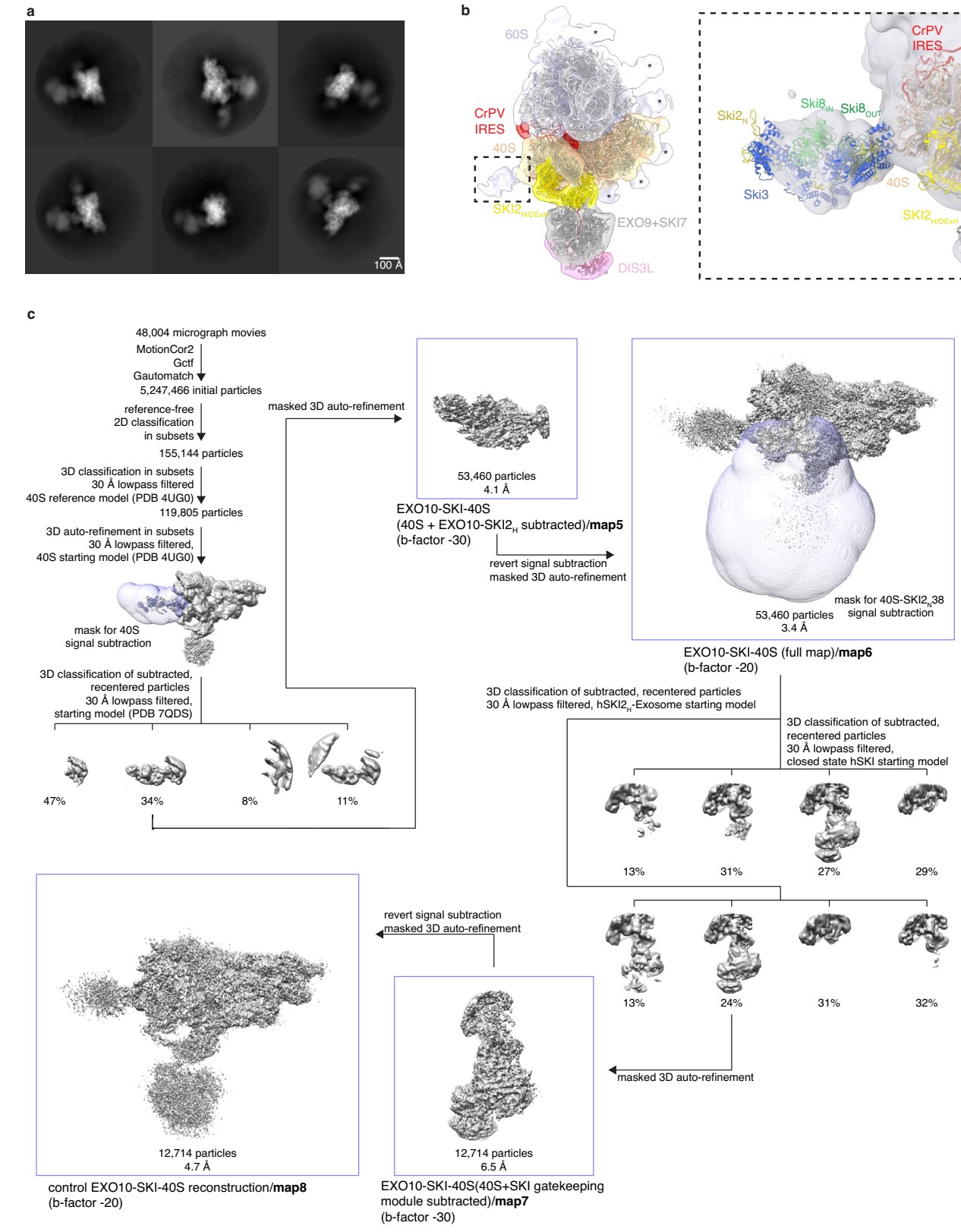

**Extended Data Fig. 6** | See next page for caption.

**Extended Data Fig. 6 | Cryo-EM data analysis of the human EXO10-SKI-40S assembly. a**, Representative 2D class averages of the human EXO10-SKI-40S assembly. These 2D class averages stem from the EXO10-SKI-80S supercomplex data recorded at 300 kV with a pixel size of 0.85 Å/pixel using a post-GIF K3 direct detector. **b**, Left panel: Low-pass-filtered full reconstruction of the EXO10-SKI-80S/map3 (cut-off 20Å) displays a density feature previously unaccounted for (black box). Right panel: indicated map area after focused 3D classification on the unaccounted density feature followed by 3D refinement (low-pass filtered to 20 Å). The structure of the EXO10-SKI-40S assembly is placed by rigid-body fitting. Colours as in Fig. 1b with SKI7 in grey. Asterisks indicate unmodelled rRNA features. **c**, Processing scheme of the EXO10-SKI-40S single-particle cryo-EM data resulting in 3D reconstructions of the EXO10-SKI-40S (full map)/map6 (at an estimated resolution of 3.4 Å) and focused reconstructions of the human SKI2$_N$38 gatekeeping module (map5; at an estimated resolution of 4.1 Å) and the EXO10-SKI assembly (map7; at an estimated resolution of 6.5 Å). These reconstructions were used to calculate a EXO10-SKI-40S composite map shown in detail in Fig. 3. Map8 is a control for the presence of SKI2$_N$38 gatekeeping module in the particles containing well-ordered EXO10-SKI. Masks used for the subtraction of partial particle signal are shown in blue.

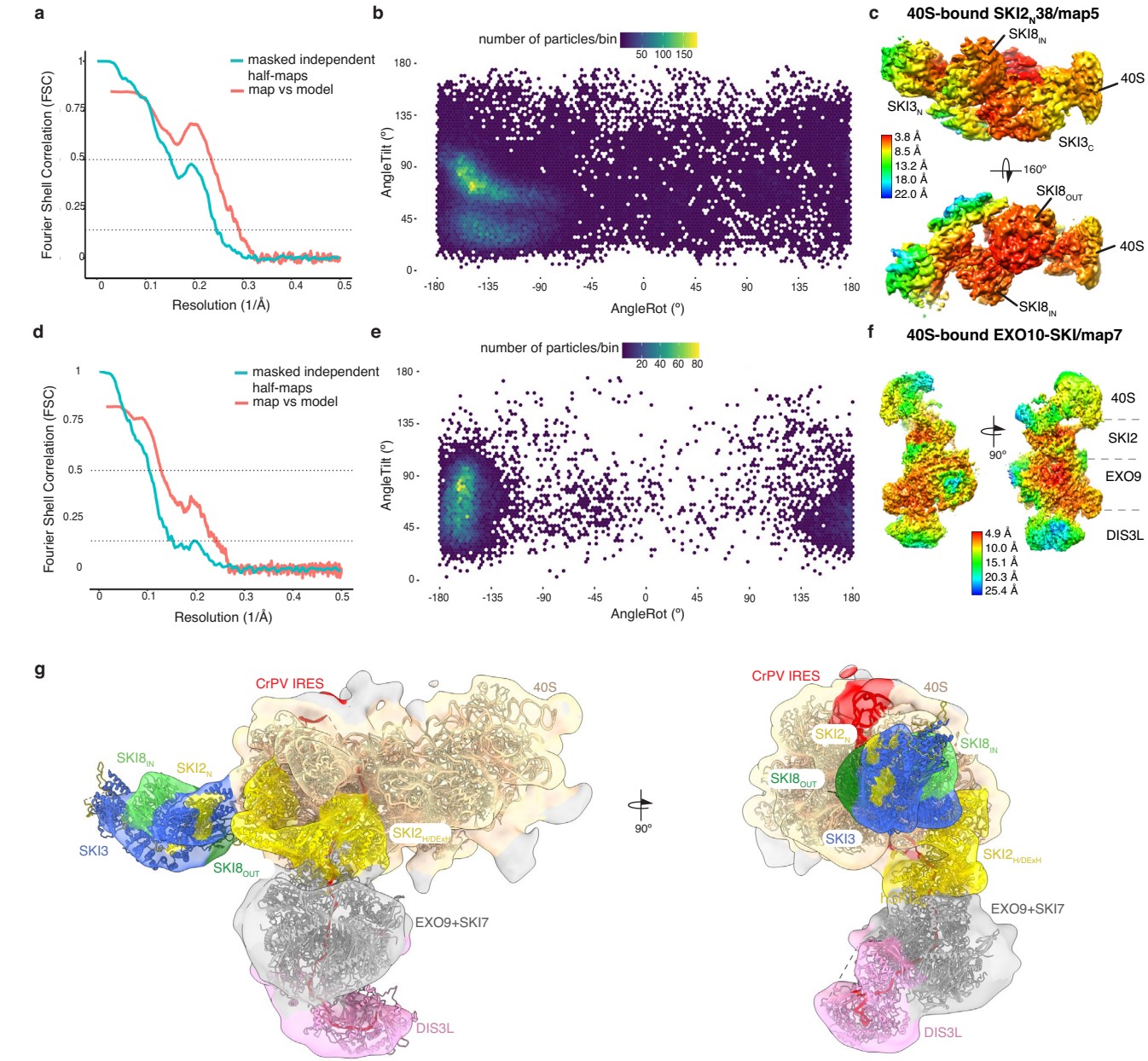

**Extended Data Fig. 7 | Quality indicators of the SPA cryo-EM reconstructions of the human EXO10-SKI-40S assembly. a**, The FSC of the masked and unmasked independent half maps for the human 40S-bound SKI2$_N$38 gatekeeping module reconstruction (map5) were calculated in the RELION 3.1 post-processing routine and the map vs model FSC using phenix.mtriage. The FSC cut-off criteria of 0.5 and 0.143 are indicated by dotted lines. **b**, Angular sampling distributions of the human 40S-bound SKI2$_N$38 gatekeeping module reconstruction (map5). Sampling angle data were plotted in 3° by 3° bins and sampling bins coloured according to particle number with yellow indicating more and blue fewer particles. **c**, Local resolution estimate of the reconstruction focused on the 40S-bound SKI2$_N$38 gatekeeping module (map5) with red indicating areas where the local resolution is estimated to be highest. Selected areas are labelled. **d**, The FSC of the masked and unmasked independent half

maps for the human 40S-bound EXO10-SKI reconstruction (map7) were calculated in the RELION 3.1 post-processing routine and the map vs model FSC using phenix.mtriage. The FSC cut-off criteria of 0.5 and 0.143 are indicated by dotted lines. **e**, Angular sampling distributions of the human 40S-bound EXO10-SKI reconstruction (map7). Sampling angle data were plotted in 3° by 3° bins and sampling bins coloured according to particle number with yellow indicating more and blue fewer particles. **f**, Local resolution estimate of the reconstruction focused on the 40S-bound EXO10-SKI assembly (map7) with red indicating areas where the local resolution is estimated to be highest. Selected areas are labelled. **g**, Low-pass-filtered control EXO10-SKI-40S reconstruction (map8; cut-off 20Å) in two orientations displays well-ordered density for 40S, SKI2$_N$38 gatekeeping module and EXO10-SKI. Colours as in Fig. 3c.

**Extended Data Fig. 8 | Structural features of the interactions between SKI7, SKI238, the exosome and the ribosome. a**, Structural superposition of the human EXO10-SKI-40S assembly on a stalled disome (PDB:7QVP)[33]. Stalled 80S ribosome and EXO10-SKI coloured similar to Fig. 1b, collided ribosome in grey, SKI2$_N$38 gatekeeping module analogues to Fig. 3a and the RNA in red. The disome unit is shown in surface representation, SKI238 and the cytosolic exosome in cartoon representation. **b**, Detailed view of the potential contact sites between human SKI2$_N$38 gatekeeping module and the 40S subunits of the stalled (grey) and the collided (sand) ribosome. Based on the superposition, the ribosomal proteins eS31 (blue), eS12 (orange) and helix H33 of the 18S rRNA

of the collided ribosome were identified as potential interactions sites for SKI8$_{IN}$. In contrast, SKI2$_H$ interacts with its RecA2 with uS3, uS12, eS10, and 18S rRNA helix 16 and with its SKI2$_{arch}$ with uS3, uS10, and 18S rRNA helix 41. **c**, Detailed view of the superposition of the SKI2$_N$38 gatekeeping module and the SKI2$_H$ helicase module on the collided disome structure (PDB:7QVP)[33], focusing on known ubiquitination sites on the 40S subunits. Purple spheres indicated the approximate ubiquitination sites of the ribosomal proteins uS3[54], eS10[55] and uS10[55]. For eS10 and uS10, the ubiquitination sites are in structurally flexible regions and not modelled. Therefore, the closest proximal residues are highlighted as spheres.

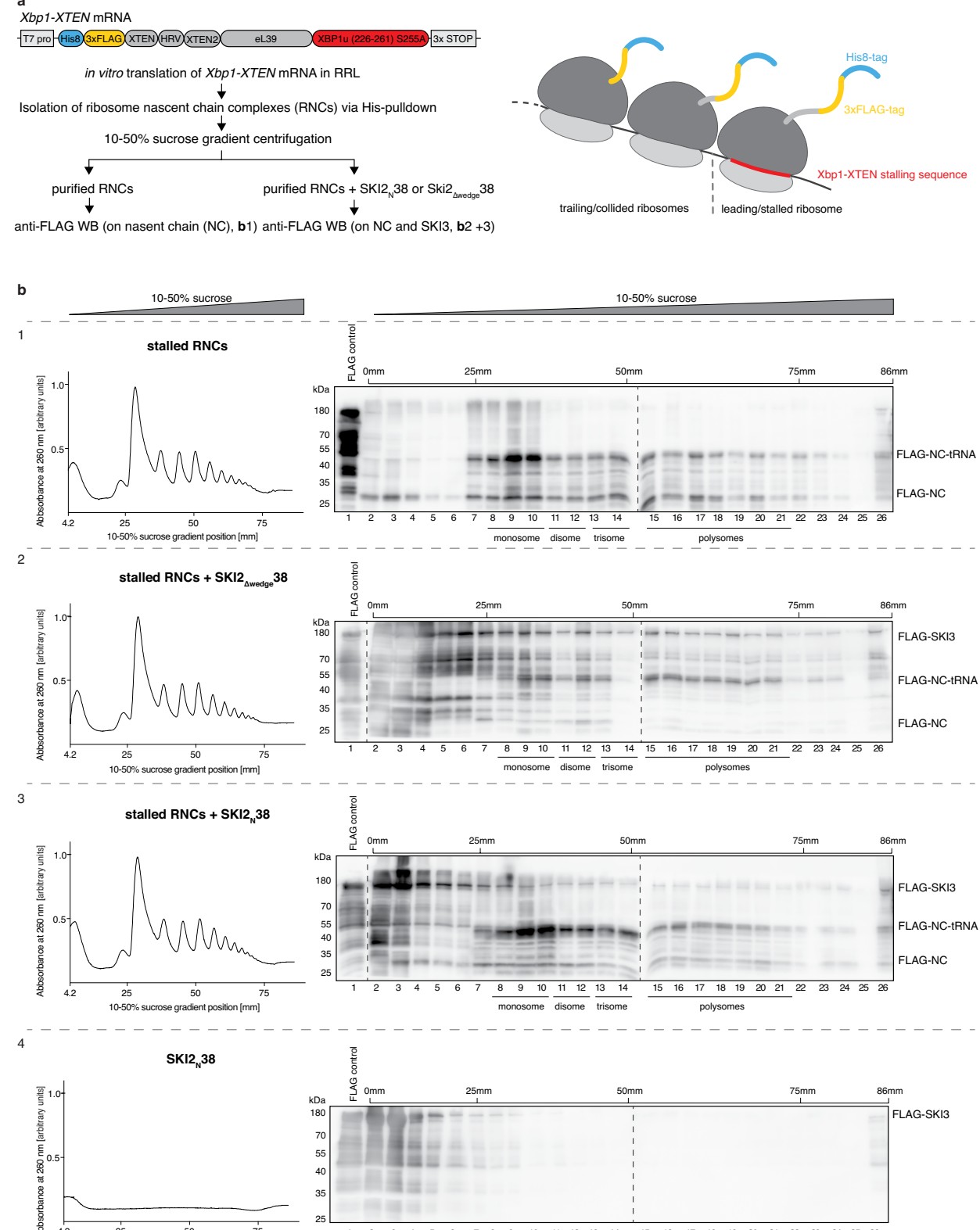

**Extended Data Fig. 9 | Interactions between collided disomes and the SKI complex. a**, Schematic of the in vitro translation and isolation protocol of ribosome nascent chain complex (RNC) using the *Xbp1-XTEN* mRNA in rabbit reticulocyte lysate (RRL). The cartoon on the right illustrates the stalling/colliding of the RNCs on the *Xbp1-XTEN* mRNA. Note that the RNCs contain both a 3xFLAG tag (used in Western blots here) and an 8xHis tag (used for purification purposes here) on the nascent chain. **b**, 10–50% sucrose gradient profiles (panel on the left) and anti-FLAG Western blots (panel on the right) of the gradient fractions of stalled RNCs only (b1), *Xbp1-XTEN* stalled RNCs with SKI2$_{\Delta wedge}$38 (b2), *Xbp1-XTEN* stalled RNCs with SKI2$_N$38 gatekeeping module (b3) and SKI2$_N$38 gatekeeping module only (b4). Flag positive control is SKI2$_N$38 gatekeeping module in panels b1, b3 and b4 as well as SKI2$_{\Delta wedge}$38 in b2. Both SKI3 and the nascent chain are Flag-tagged.

## Extended Data Table 1 | Cryo-EM data collection summary, processing statistics and model quality indicators

| | Map1/ EXO10-SKI-80S (80S subtracted) (EMDB-51133) (PDB 9G8N) | Map2/ EXO10-SKI-80S (60S subtracted) (EMDB-51139) | Map3/ EXO10-SKI-80S (full map) (EMDB-51132) (PDB 9G8M) | Map5/ EXO10-SKI-40S (EXO10-SKI2$_H$ + 40S subtracted) (EMDB-51136) (PDB 9G8Q) | Map6/ EXO10-SKI-40S (full map) (EMDB-51134) (PDB 9G8O) | Map7/ EXO10-SKI-40S (40S + SKI gatekeeping module subtracted) (EMDB-51135) (PDB 9G8P) | Map8/ Control EXO10-SKI-40S (EMDB-51135) | Map4/ SKI7-SKI238 (EMDB-51137) (PDB 9G8R) |
|---|---|---|---|---|---|---|---|---|
| **Data collection and processing** | | | | | | | | |
| Microscope | | | | Titan Krios | | | | Titan Krios |
| Magnification | | | | 105000 | | | | 105000 |
| Voltage (kV) | | | | 300 | | | | 300 |
| Electron exposure (e–/Å$^2$) | | | | 64.2 | | | | 67.8 |
| Defocus range (μm) | | | | 0.6 – 2.4 | | | | 0.6 – 2.4 |
| Pixel size (Å) | | | | 0.8512 | | | | 0.8512 |
| Symmetry imposed | | | | C1 | | | | C1 |
| Initial particle images (no.) | | | | 5,247,466 | | | | 7,323,297 |
| Final particle images (no.) | | 79353 | | | 53,460 | | 12,714 | 151,860 |
| Map resolution (Å) FSC threshold | 3.7 | 3.3 | 3.3 | 4.1 | 3.4 | 7 | 4.7 | 3.4 |
| Map resolution range (Å) | 3.3 - 8.7 | - | - | 3.8 - 18 | - | 4.9 – 20.3 | - | 2.8 – 7.8 |
| **Refinement** | | | | | | | | |
| Initial model used (PDB code) | 6D6Q | | 4UG0 | 7QDS | 4UG0 | 6D6Q | | 7QDS |
| Model resolution (Å) | 3.1 | | 2.8 | 3.4 | 3.1 | 4.1 | | 2.8 |
| FSC threshold | 0.143 | | 0.143 | 0.143 | 0.143 | 0.143 | | 0.143 |
| Map sharpening *B* factor (Å$^2$) (determined/applied) | -110.8/-30 | -75.70/-20 | -71.7/-20 | -134.4/-30 | -68.9/-20 | -128.3/-30 | -92.5/-20 | -99.4/-99.4 |
| Model composition | | | | | | | | |
| Non-hydrogen atoms | 34406 | | 251093 | 13313 | 125223 | 34406 | | 14837 |
| Protein residues | 4304 | | 15755 | 1707 | 10899 | 4304 | | 1900 |
| Ligands | 0 | | 232 MG 6 ZN | 0 | 0 | 0 | | 0 |
| *B* factors (Å$^2$) | | | | | | | | |
| Protein | 134.80 | | 178.38 | 131.11 | 473.83 | 307.91 | | 69.26 |
| Ligand | | | 93.86 | | | | | |
| R.m.s. deviations | | | | | | | | |
| Bond lengths (Å) | 0.003 | | 0.005 | 0.004 | 0.005 | 0.004 | | 0.003 |
| Bond angles (°) | 0.800 | | 0.775 | 0.898 | 0.735 | 0.958 | | 0.838 |
| Validation | | | | | | | | |
| MolProbity score | 1.83 | | 2.13 | 1.85 | 2.11 | 1.82 | | 1.78 |
| Clashscore | 10.95 | | 11.80 | 13.33 | 12.03 | 16.51 | | 10.72 |
| Poor rotamers (%) | 0.00 | | 0.36 | 0.00 | 0.01 | 0.00 | | 0.00 |
| Ramachandran plot | | | | | | | | |
| Favored (%) | 95.99 | | 90.03 | 96.58 | 91.24 | 97.51 | | 96.50 |
| Allowed (%) | 3.97 | | 9.10 | 3.24 | 8.37 | 2.44 | | 3.34 |
| Disallowed (%) | 0.05 | | 0.86 | 0.18 | 0.39 | 0.05 | | 0.016 |

[a]According to the FSC cut-off criterion of 0.143 defined in ref. 46.
[b]According to the map-versus-model correlation coefficient definitions in ref. 56.
*Cross-correlation coefficients between composite maps and model.

# Reporting Summary

## Statistics

For all statistical analyses, confirm that the following items are present in the figure legend, table legend, main text, or Methods section.

| n/a | Confirmed | |
|---|---|---|
| ☐ | ☒ | The exact sample size (*n*) for each experimental group/condition, given as a discrete number and unit of measurement |
| ☐ | ☒ | A statement on whether measurements were taken from distinct samples or whether the same sample was measured repeatedly |
| ☒ | ☐ | The statistical test(s) used AND whether they are one- or two-sided<br>*Only common tests should be described solely by name; describe more complex techniques in the Methods section.* |
| ☒ | ☐ | A description of all covariates tested |
| ☒ | ☐ | A description of any assumptions or corrections, such as tests of normality and adjustment for multiple comparisons |
| ☐ | ☒ | A full description of the statistical parameters including central tendency (e.g. means) or other basic estimates (e.g. regression coefficient) AND variation (e.g. standard deviation) or associated estimates of uncertainty (e.g. confidence intervals) |
| ☒ | ☐ | For null hypothesis testing, the test statistic (e.g. *F*, *t*, *r*) with confidence intervals, effect sizes, degrees of freedom and *P* value noted<br>*Give P values as exact values whenever suitable.* |
| ☒ | ☐ | For Bayesian analysis, information on the choice of priors and Markov chain Monte Carlo settings |
| ☒ | ☐ | For hierarchical and complex designs, identification of the appropriate level for tests and full reporting of outcomes |
| ☒ | ☐ | Estimates of effect sizes (e.g. Cohen's *d*, Pearson's *r*), indicating how they were calculated |

*Our web collection on statistics for biologists contains articles on many of the points above.*

## Software and code

Policy information about availability of computer code

| Data collection | SerialEM 4.0; Digital Micrograph 3.32; Focus 1.1.0 |
|---|---|
| Data analysis | Motioncor2; GCTF 1.06; Gautomatch 0.56; RELION 3.1; ChimeraX 1.6.1; PHENIX 1.20; Coot 0.8.9; ImageJ |

For manuscripts utilizing custom algorithms or software that are central to the research but not yet described in published literature, software must be made available to editors and reviewers. We strongly encourage code deposition in a community repository (e.g. GitHub). See the Nature Portfolio guidelines for submitting code & software for further information.

## Data

Policy information about availability of data

All manuscripts must include a data availability statement. This statement should provide the following information, where applicable:
- Accession codes, unique identifiers, or web links for publicly available datasets
- A description of any restrictions on data availability
- For clinical datasets or third party data, please ensure that the statement adheres to our policy

Cryo-EM density maps that support the findings in this study have been deposited in the Electron Microscopy Data Bank (EMDB) and the Protein Data Bank (PDB) under the accession numbers: Map1/EXO10-SKI-80S (80S subtracted) EMDB: 51133, PDB: 9G8N; Map2/EXO10SKI-80S (60S subtracted) EMDB: 51139; Map3/EXO10-SKI-80S (full map) EMDB: 51132, PDB: (9G8M); Map4/SKI7-SKI238 EMDB: 51137, PDB: 9G8R; Map5/ EXO10-SKI-40S (EXO10-SKI2H + 40S subtracted) EMDB: 51136, PDB: 9G8Q; Map6/EXO10-SKI-40S (full map) EMDB-51134, PDB; 9G8O; Map7/EXO10-SKI-40S (40S + SKI gatekeeping module subtracted); EMDB 51135, PDB: 9G8;

Map8/Control EXO10-SKI-40S EMDB-51135. All other data are available within the main text or the Extended Data. Raw data and source images are available in the supplementary information.

# Research involving human participants, their data, or biological material

Policy information about studies with [human participants or human data](). See also policy information about [sex, gender (identity/presentation), and sexual orientation]() and [race, ethnicity and racism]().

| | |
|---|---|
| Reporting on sex and gender | These data were not collected in this study |
| Reporting on race, ethnicity, or other socially relevant groupings | These data were not collected in this study. |
| Population characteristics | These data were not collected in this study. |
| Recruitment | Participants were not recruited for this study. |
| Ethics oversight | No approval number was required for this study. |

Note that full information on the approval of the study protocol must also be provided in the manuscript.

# Field-specific reporting

Please select the one below that is the best fit for your research. If you are not sure, read the appropriate sections before making your selection.

☒ Life sciences ☐ Behavioural & social sciences ☐ Ecological, evolutionary & environmental sciences

For a reference copy of the document with all sections, see [nature.com/documents/nr-reporting-summary-flat.pdf]()

# Life sciences study design

All studies must disclose on these points even when the disclosure is negative.

| | |
|---|---|
| Sample size | No sample sizes were calculated. CryoEM data were collected over multiple days to yield a sufficient amount of particles and to obtain high resolution 3D reconstructions of the complexes of interest. The number of technical replicates in the extended data figure 1f was chosen based on the minimum amount required for statistical verification that the RNA to be used in structural studies was an appropriate substrate, although no calculations were performed. |
| Data exclusions | During cryoEM data processing, particles were excluded during 2D and 3D classification when missing clear secondary structure features in 2D class averages or 3D reconstructions. |
| Replication | Western blots, pulldown assays, and degradation assays were performed in independent triplicates to confirm similar results. Raw images and source data are shown in Supplementary. |
| Randomization | Randomization was not relevant to this study because we did not include human subjects or animals. |
| Blinding | Blinding was not relevant to this study because we did not include human subjects or animals. |

# Reporting for specific materials, systems and methods

We require information from authors about some types of materials, experimental systems and methods used in many studies. Here, indicate whether each material, system or method listed is relevant to your study. If you are not sure if a list item applies to your research, read the appropriate section before selecting a response.

## Materials & experimental systems

| n/a | Involved in the study |
|---|---|
| ☐ | ☒ Antibodies |
| ☐ | ☒ Eukaryotic cell lines |
| ☒ | ☐ Palaeontology and archaeology |
| ☒ | ☐ Animals and other organisms |
| ☒ | ☐ Clinical data |
| ☒ | ☐ Dual use research of concern |
| ☒ | ☐ Plants |

## Methods

| n/a | Involved in the study |
|---|---|
| ☒ | ☐ ChIP-seq |
| ☒ | ☐ Flow cytometry |
| ☒ | ☐ MRI-based neuroimaging |

## Antibodies

| | |
|---|---|
| Antibodies used | Monoclonal ANTI-FLAG clone M2 mouse antibody (Sigma-Aldrich, Cat. No. F3165); Polyclonal anti-mouse HRP-coupled antibody (Bio-Rad, Cat. No. 172-1011) |
| Validation | Monoclonal ANTI-FLAG antibody was validated against purified SKI-complexes, where the SKI3 subunit was FLAG tagged ( Extended Data Fig. 9) |

## Eukaryotic cell lines

Policy information about cell lines and Sex and Gender in Research

| | |
|---|---|
| Cell line source(s) | HEK293T - ATCC; Hi5 - Invitrogen (Thermo Fisher Scientific) |
| Authentication | None of the cell lines were authenticated. |
| Mycoplasma contamination | All cell lines tested negative for mycoplasma contamination based on regular testing. |
| Commonly misidentified lines (See ICLAC register) | No commonly misidentified cell lines were used in this study. |

## Plants

| | |
|---|---|
| Seed stocks | No seed stocks were used in this study. |
| Novel plant genotypes | No novel plant genotypes were generated in this study. |
| Authentication | This is not applicable to this study. |

