## [Peer Review File · Nature]

Manuscript Title: Structural basis of mRNA decay by the human exosome-ribosome supercomplex

Reviewer Comments & Author Rebuttals

Reviewer Reports on the Initial Version:

Referees' comments:

Referee #1 (Remarks to the Author):

The manuscript by Kögel et al. describes the structures of the exosome-ribosome-SKI supercomplex, extending their prior investigations into various functional states of the SKI complex in conjunction with ribosomes and substrate RNA based on CRPV IRES. The authors achieved structural resolution ranging from 3.2 to 8.7 Å, allowing for the integration of known protein structures into the EM density map. They demonstrate that HBS1L3/SKI7 recruits EXO10 towards an active, ribosome-bound SKI238 complex through interactions with SKI3. This research significantly advances our understanding by elucidating the direct interplay between ribosomes, the SKI complex, and the exosome within mRNA degradation pathways, which is pertinent to both yeast and human systems. While this represents a substantial step forward, the conceptual advance on functional understandings remains unclear, especially concerning the supercomplex's role in the collided ribosomes.

Major Comments:

The manuscript would benefit from a deeper exploration into why modeling the supercomplex in association with both the monosome and the disome is functionally relevant, as it is currently presented as a major conclusion figure. Specifically, it should address whether interaction with the disome aids in the transition of SKI238 to an open conformation, thereby promoting mRNA degradation and resolution of ribosome collisions. Providing biochemical or structural evidence to support this model would greatly enhance the manuscript's impact.

The discrepancy in the disorder levels of the EM density for the SKI gatekeeping module and SKI7SKI in relation to the 80S versus the 40S ribosome warrants further analysis. The authors should offer an explanation for this observation and validate the positioning of the gatekeeping module within the additional density observed in the 80S ribosome complex as shown in Extended Data Figure 5b. Additionally, particle subtraction and focused refinement for the 40S head and body in the 80S complex could potentially enhance map clarity and the visibility of the gatekeeping module.

A comparison between the previously determined structure of the SKI7-GTP complex by Conti and colleagues and its configuration within this supercomplex could provide valuable insights, especially in the context of SKI7's role in recruiting the SKI238 and EXO10 complexes. Further, an extended data figure featuring a detailed sequence alignment between yeast Ski7 and HSB1L3, and a comparative

analysis between human and yeast SKI7 in their respective complexes, would enhance the manuscript's depth.

Minor Points:

The visualization of mRNA tracing through the ribosome and SKI complex into the exoribonuclease complex in Figure 1 could be improved. A revised figure highlighting the cryo-EM density for the mRNA and its interaction regions would better support the authors' claims.

The terms “eluate” and “precipitate” are interchangeably used to describe the same assay results in different figures. Using one term throughout the manuscript would prevent confusion.

For clarity, the DExH core should be labeled in Figure 2a (possibly as SKI2H/DExH), or the text should consistently refer to it as SKI2H.

The presentation of the AlphaFold prediction and the corresponding EM density map for the SKI3 superhelix and SKI7 helix interaction (Fig. 3b) should be improved to validate the model, especially showing the fit of residues 451-464 of SKI7 within the SKI3 structure.

References for the role of ubiquitylation during ribosome-associated quality control as mentioned in connection to Extended Data Fig. 7c should be included.

The specific vector/plasmid used for cloning SKI7 should be named to allow for replication of results.

The manuscript should include references for PDB ID 7QVP to provide proper attribution and context for readers.

Referee #2 (Remarks to the Author):

Previous analyses, by the authors and others, have established the conformational inhibition of SKI2 within the SKI complex, the role of the activated complex in extracting the mRNA from stalled ribosomes and channeling through to the exosome lumen. Here the authors report structures and protein-protein interactions that support a model for direct channeling of the mRNA undergoing extraction from the ribosome through SKI2 the DIS3L active site. The 80S-mRNA-SKI-exosome complex is a technical tour de force. The gatekeeper module was not clearly resolved, but is convincingly mapped in a simpler complex. They also postulate SKI binding to collided ribosomes, which have emerged as key markers for translational defects. This is the most novel finding and the data presented show that this is a feasible, and probable, set of interactions. They fall short of a demonstration that this actually occurs, but this is a proposal that will stimulate the field.

Overall, the work is technically excellent. It brings previous results together and expends them into a

model that will be of wide interest.

Minor points:

1: Ext Fig. 1c: The lanes do not appear to be as indicated by at the top of the gels. I suspect that the positive and negative controls are inverted.

2: Ext Fig. 1f: The unquantified loss of Brc signal is not very convincing as evidence for activated degradation.

Referee #3 (Remarks to the Author):

In this manuscript the authors present a structure of the eukaryotic ribosome undergoing decay by the exosome. The cytoplasmic exosome is a multisubunit complex that relies on cofactors to coordinate delivery of substrates to the exonuclease Dis3. The helicase complex SKI238 associates with ribosomes. If triggered by aberrant stalling or translation termination, SKI238 extracts the 3' end of ribosome associated mRNAs and delivers them to the exosome for decay. How the exosome is recruited to the SKI238-ribosome complex has remained unknown. By analyzing proteomics data the authors hypothesized that SKI7 may function as the bridging factor between SKI238 and the exosome. To test this hypothesis, they performed binding and activity assays with recombinant proteins. They identified two important interaction regions within the C-terminus of SKI7, one which mediates SKI238 binding (SKI) and the other that mediates EXO10 binding (EXO). Next the authors reconstituted the full assembly using a CrPV IRES as the mRNA substrate and catalytically deficient EXO10 and were able to determine a cryo-EM structure of the mega complex. The structure revealed clear density for the substrate as it moves through the complex to the Dis3L active site. Interestingly the authors observed that the endonuclease domain of Dis3L suggests a structural rather than functional role. While the structure revealed important insight into the interactions formed between Ski2-Ski7-EXO10, missing from the structure was ordered density for the SKI7-Ski domain and the SKI238 complex. The authors speculated that the SKI7-Ski domain may bind to the gatekeeping complex so they determined a structure of the minimal SKI7-SKI238 complex, revealing how a small helix from SKI7 associates with SKI3. This interface was confirmed by mutagenesis. Next the authors observed that the density for the minimal SKI7-SKI238 complex shared features of poorly defined density visible on both 40S and 80S particles in their previous cryo-EM dataset. Through additional processing/modeling the authors were able to dock this module onto the ribosome structures providing details about how the TPR repeats from SKI3 associate with the head of the 40S. Finally, the authors modeled the SKI gatekeeping module onto the structure of a collided diasome making the observation that the complex could bind to both stalled and collided ribosomes, suggesting a putative mechanism of targeted recruitment. Overall, this is an impressive manuscript providing near-atomic resolution details about exosome mediated mRNA decay on the ribosome.

Specific Comments:

1. It is hard to understand the difference between the open and closed conformations of SKI238. A supplemental figure showing a side-by-side comparison would be helpful.
2. The exosome alone is a complicated machine with many subunits. In general, the authors did a great job of not introducing too many protein names but noted below are a few examples that could use some additional details in the text to help readers follow.
 - a. Line 151 – Since several subunits are highlighted in detail it would be helpful to list in () the names of the 6 PH-like proteins and the three S1/KH cap subunits.
 - b. Line 170 – Define that a PIN domain is an endonuclease domain and RNase II is an exonuclease domain.
 - c. Line 184 (and several other places) “Gatekeeping Module”. I suggest referring to this as the “SKI2N38 Gatekeeping Module” throughout the manuscript to remind readers it includes portions of all three proteins.
3. If space allows it would be nice to show a side-by-side comparison of the PIN domains from Dis3 and Dis3L illustrating why the authors believe Dis3L is not active. Is there any biochemical evidence to support this? Could the inactive conformation be predicted from the alphafold model alone?
4. In Fig 3A it is unclear what TPR30A, TPR30B, and TPR29A is referring to. Please define in the figure legend.
5. I’m curious about the THES disease mutations. Is there any evidence to suggest these mutations weaken the association with the ribosome or other Ski subunits? It is hard to tell in the figure, but these mutations also appear to be close to the interfaces with Ski8 and Ski2.
6. In the cryo-EM data processing workflows it would be helpful to add some labels on the local resolution maps to help readers understand what is what.

Author Rebuttals to Initial Comments:

We wish to thank the Reviewers for their thoughtful suggestions and constructive criticisms, and also for their appreciative comments on our work and manuscript. We addressed all the comments and incorporated the corresponding changes and additional data in the text and Figures, as detailed below. Additions to the manuscript, implemented to address the Referees' comments, have been highlighted in blue to facilitate the re-review. To accommodate the additional text, we streamlined the more detailed segments of the original version, which we think also improved the focus of the narrative. We would also like to bring to the attention of the Reviewers that due to the addition of new Extended Data Fig. 5, the following supplementary figures have been renumbered (i.e., Extended Data Fig. 5, 6, and 7 in the original manuscript correspond to Extended Data Fig. 6, 7, and 8 in the revised manuscript).

Referees' comments:**Referee #1 (Remarks to the Author):**

The manuscript by Kögel et al. describes the structures of the exosome-ribosome-SKI supercomplex, extending their prior investigations into various functional states of the SKI complex in conjunction with ribosomes and substrate RNA based on CRPV IRES. The authors achieved structural resolution ranging from 3.2 to 8.7 Å, allowing for the integration of known protein structures into the EM density map. They demonstrate that HBS1L3/SKI7 recruits EXO10 towards an active, ribosome-bound SKI238 complex through interactions with SKI3. This research significantly advances our understanding by elucidating the direct interplay between ribosomes, the SKI complex, and the exosome within mRNA degradation pathways, which is pertinent to both yeast and human systems. While this represents a substantial step

forward, the conceptual advance on functional understandings remains unclear, especially concerning the supercomplex's role in the collided ribosomes.

As we hope the Reviewer will appreciate, we took the constructive criticism in our stride to improve the manuscript and more effectively convey the significance of the mechanistic insights we report.

Major Comments

The manuscript would benefit from a deeper exploration into why modeling the supercomplex in association with both the monosome and the disome is functionally relevant, as it is currently presented as a major conclusion figure. Specifically, it should address whether interaction with the disome aids in the transition of SKI238 to an open conformation, thereby promoting mRNA degradation and resolution of ribosome collisions. Providing biochemical or structural evidence to support this model would greatly enhance the manuscript's impact.

We agree with the Reviewer that the model in Fig. 4 in the original manuscript may have appeared as an isolated finding in the context of the other Figures in the main text. We have now improved Figure 4 and the text in several ways, and importantly added new biochemical evidence in support of the disome collision model.

First, we refer to and discuss previous data showing that SKI2 becomes enriched on transcripts upon ribosome stalling events in cells, as assessed by crosslinking and profiling data (Tuck et al. 2020). While at the time these results from cell-based assays could not be interpreted in the framework of mechanistic hypotheses, with hindsight they corroborate the functional significance of the insights we obtained from structural and biochemical data.

Second, we created a new figure (now Fig. 4a) depicting the superposition of the coordinates between a previous disome structure (Narita, M., et al., 2022) and our cytoplasmic exosome-bound structure presented in Fig. 3c to show the context for the model schematized in Fig. 4c (updated to also address point 1 of Reviewer 3).

Last but not least, we now support the model with biochemical evidence, in new Fig. 4b and new Extended Data Fig. 9. As stated in the text:

‘To test the structure-based hypothesis that the open state conformation of SKI238 may recognize collided disomes, we conducted polysome profiling experiments by inducing translation stalling in the rabbit reticulocyte lysate (RRL) *in vitro* system with the previously described *Xbp1-XTEN* mRNA. Ribosome nascent chain complexes (RNCs) were affinity

purified via the His-tag encoded in the reporter and were incubated in three different conditions (either as a control RNCs-only sample, or with recombinant SKI2 Δ wedge38 mutant complex previously shown to favor the open-state conformation), or with recombinant SKI2_N38 gatekeeping module that lacks the 80S-SKI2_H interaction) before being subjected to ultracentrifugation on a sucrose density gradient (Fig. 4b, Extended Data Fig. 9a). The FLAG-tag encoded in the reporter (resulting in a ~ 45 kDa peptidyl-tRNA product upon translation elongation stalling) and in recombinant SKI3 (~ 175 kDa) allowed to then analyze the polysome profile fractions by anti-FLAG Western blotting (Fig. 4b and Extended Fig. 9b). These biochemical analyses demonstrate that both the open-state SKI2 Δ wedge38 complex and the SKI2_N38 gatekeeping module can be enriched in the disome fractions from sucrose gradients of stalled RNCs (Fig. 4b and Extended Fig. 9b) suggesting a direct interaction.'

Regarding the specific sequence of events, we deliberately chose not to depict their possible order. The reason for this is that while we obtain the first structural snapshot in this process, we do not have information on dynamics. Dissecting the detailed sequence of events will require a different set of sophisticated experimental approaches, such as single-molecule FRET, and is beyond the scope of the current manuscript. However, we do believe that the results we report here will stimulate the field and inspire new perspectives in the study of co-translational surveillance pathways.

The discrepancy in the disorder levels of the EM density for the SKI gatekeeping module and SKI7SKI in relation to the 80S versus the 40S ribosome warrants further analysis. The authors should offer an explanation for this observation and validate the positioning of the gatekeeping module within the additional density observed in the 80S ribosome complex as shown in Extended Data Figure 5b. Additionally, particle subtraction and focused refinement for the 40S head and body in the 80S complex could potentially enhance map clarity and the visibility of the gatekeeping module.

As suggested by the Reviewer, we have now improved the visibility of the gatekeeping module in the 80S reconstruction (new Extended Data Figure 5b). Although we tried the approach suggested by the Reviewer (*particle subtraction and focused refinement for the 40S head and body in the 80S complex*), the results we obtained did not improve the density of the 80S-bound SKI2_N38 gatekeeping module beyond what we had shown in the original Extended Data Figure 5b and were thus not satisfactory. We therefore proceeded with a different strategy. We performed focused 3D classification using a spherical mask located around the identified

gatekeeping-binding site on the 40S. The resulting particle stack was used for 3D refinement resulting in a 3D reconstruction of the 80S ribosome with better density for the SKI2 helicase as well as the SKI2_{N38} gatekeeping module. A low-pass filtered map of this assembly allowed us to confidently place the model of SKI2_{N38} within the density of the 80S reconstruction by rigid body fitting. Upon comparing the 80S and 40S reconstructions, we do not detect significant differences in the overall orientation and positioning of the gatekeeping module.

A comparison between the previously determined structure of the SKI7-GTP complex by Conti and colleagues and its configuration within this supercomplex could provide valuable insights, especially in the context of SKI7's role in recruiting the SKI238 and EXO10 complexes.

We believe that the Reviewer refers to the structure of the yeast Ski7 GTP-binding domain reported in Kowalinski et al., 2015. If so, it seems we did not make it clear enough in our original submission that human SKI7 does not contain this domain. We now included a schematic that allows the reader to compare the domain architecture of yeast Ski7 and human SKI7 (new Extended Data Figure 1a). This panel figure should now allow the readership to better appreciate that they are different proteins (with human SKI7 being an alternatively spliced isoform of *HBSIL* lacking the GTP-binding domain), despite having converged to serve a similar function.

Further, an extended data figure featuring a detailed sequence alignment between yeast Ski7 and HSB1L3, and a comparative analysis between human and yeast SKI7 in their respective complexes, would enhance the manuscript's depth.

We added a supplementary figure (new Extended Data Fig. 5) with sequence alignments of the SKI-binding domain of SKI7 in several species, as well as a comparison of the structures from the human complex reported in this manuscript and the structure of the yeast complex reported in Keidel et al., 2023. These new figures should facilitate a better appreciation of the differences in the individual binding determinants. For the sake of completion, in Supplementary material we also now show an additional helix of human SKI7 (SKI7-H1) that can be predicted by AlphaFold, but for which density in our EM data was not sufficiently clear and for which we did not observe a significant contribution to binding. Helix H2 corresponds to the well-defined helix in the map and is the major binding determinant.

We also now clarify in the text that, in contrast to the divergence of the SKI-binding domain, the exosome-binding domain of SKI7 follows a similar binding path as the yeast orthologue. The similarity in stretches of the exosome-binding sequence is indeed what enabled the original

identification of human SKI7 (Kowalinski et al., 2016, Kalisiak et al., 2016). A sequence alignment of the exosome-binding domain is in Kowalinski et al., 2016. Considering the overall length of the current manuscript, we believe that a more detailed comparative analysis between yeast and human complexes may be better suited for a review.

Minor Points

The visualization of mRNA tracing through the ribosome and SKI complex into the exoribonuclease complex in Figure 1 could be improved. A revised figure highlighting the cryo-EM density for the mRNA and its interaction regions would better support the authors' claims.

In the revised manuscript, we now include a zoom-in view highlighting the cryo-EM density for mRNA (new Extended Fig. 2h).

The terms “eluate” and “precipitate” are interchangeably used to describe the same assay results in different figures. Using one term throughout the manuscript would prevent confusion.

We now only use the term ‘precipitate’ throughout the manuscript.

For clarity, the DExH core should be labeled in Figure 2a (possibly as SKI2H/DExH), or the text should consistently refer to it as SKI2H.

We revised the figures to include the term DExH.

The presentation of the AlphaFold prediction and the corresponding EM density map for the SKI3 superhelix and SKI7 helix interaction (Fig. 3b) should be improved to validate the model, especially showing the fit of residues 451-464 of SKI7 within the SKI3 structure.

The presentation of the AlphaFold prediction in the manuscript was actually the clearest view among those we tried. We would like to emphasize that we validated the model experimentally by pull-down assays with recombinant proteins and mutants (Extended Data Fig. 4h).

References for the role of ubiquitylation during ribosome-associated quality control as mentioned in connection to Extended Data Fig. 7c should be included.

We now added the reference corresponding to PDB 7QVP (Narita et al. 2022) in the legend of the Extended Data where this structure is shown (now numbered 8c instead of 7c). We also

included this reference as well as Chandrasekaran et al., 2018, Juszkievicz et al. 2018 and Simms et al. 2017 in the main text to support the associated statement(s).

The specific vector/plasmid used for cloning SKI7 should be named to allow for replication of results.

The plasmid we used is a MPIB in-house *E.coli* expression plasmid named pEC-A-6xHis-TrxA-3C-hSKI7-eGFP-TwinStrepII which confers Ampicillin resistance.

The manuscript should include references for PDB ID 7QVP to provide proper attribution and context for readers.

We apologize for this oversight and have now added the appropriate reference.

Referee #2 (Remarks to the Author):

Previous analyses, by the authors and others, have established the conformational inhibition of SKI2 within the SKI complex, the role of the activated complex in extracting the mRNA from stalled ribosomes and channeling through to the exosome lumen. Here the authors report structures and protein-protein interactions that support a model for direct channeling of the mRNA undergoing extraction from the ribosome through SKI2 the DIS3L active site. The 80S-mRNA-SKI-exosome complex is a technical tour de force. The gatekeeper module was not clearly resolved, but is convincingly mapped in a simpler complex. They also postulate SKI binding to collided ribosomes, which have emerged as key markers for translational defects. This is the most novel finding and the data presented show that this is a feasible, and probable, set of interactions. They fall short of a demonstration that this actually occurs, but this is a proposal that will stimulate the field. Overall, the work is technically excellent. It brings previous results together and expends them into a model that will be of wide interest.

Minor points

1: Ext Fig. 1c: The lanes do not appear to be as indicated by at the top of the gels. I suspect that the positive and negative controls are inverted.

We thank the Reviewer for pointing out this mistake. The labeling has been corrected.

2: Ext Fig. 1f: The unquantified loss of Brc signal is not very convincing as evidence for activated degradation.

This experiment simply served as a control for us to confirm that the RNA substrate we intended to use for the structural analyses was suitable for our experimental strategy. However, we agree with the Reviewer that the single experiment in the original submission was in general not sufficiently convincing to the readership. We have now carried out this experiment in triplicates and performed a densitometric quantitation of each of the degradation reactions. This new data panel (new Extended Data Fig. 1f, central panel) substantiates our conclusion that the RNA substrate we used for the structural analyses can indeed be actively degraded, and also corroborates previous data (Zinoviev et al., 2020) that the cytoplasmic exosome can extract and degrade RNA in the presence of ATP but not ADP.

Referee #3 (Remarks to the Author):

In this manuscript the authors present a structure of the eukaryotic ribosome undergoing decay by the exosome. The cytoplasmic exosome is a multisubunit complex that relies on cofactors to coordinate delivery of substrates to the exonuclease Dis3. The helicase complex SKI238 associates with ribosomes. If triggered by aberrant stalling or translation termination, SKI238 extracts the 3' end of ribosome associated mRNAs and delivers them to the exosome for decay. How the exosome is recruited to the SKI238-ribosome complex has remained unknown. By analyzing proteomics data the authors hypothesized that SKI7 may function as the bridging factor between SKI238 and the exosome. To test this hypothesis, they performed binding and activity assays with recombinant proteins. They identified two important interaction regions within the C-terminus of SKI7, one which mediates SKI238 binding (SKI) and the other that mediates EXO10 binding (EXO). Next the authors reconstituted the full assembly using a CrPV IRES as the mRNA substrate and catalytically deficient EXO10 and were able to determine a cryo-EM structure of the mega complex. The structure revealed clear density for the substrate as it moves through the complex to the Dis3L active site. Interestingly the authors observed that the endonuclease domain of Dis3L suggests a structural rather than functional role. While the structure revealed important insight into the interactions formed between Ski2-Ski7-EXO10, missing from the structure was ordered density for the SKI7-Ski domain and the SKI238

complex. The authors speculated that the SKI7-Ski domain may bind to the gatekeeping complex so they determined a structure of the minimal SKI7-SKI238 complex, revealing how a small helix from SKI7 associates with SKI3. This interface was confirmed by mutagenesis. Next the authors observed that the density for the minimal SKI7-SKI238 complex shared features of poorly defined density visible on both 40S and 80S particles in their previous cryo-EM dataset. Through additional processing/modeling the authors were able to dock this module onto the ribosome structures providing details about how the TPR repeats from SKI3 associate with the head of the 40S. Finally, the authors modeled the SKI gatekeeping module onto the structure of a collided disome making the observation that the complex could bind to both stalled and collided ribosomes, suggesting a putative mechanism of targeted recruitment. Overall, this is an impressive manuscript providing near-atomic resolution details about exosome mediated mRNA decay on the ribosome.

Specific Comments

1. It is hard to understand the difference between the open and closed conformations of SKI238. A supplemental figure showing a side-by-side comparison would be helpful.

We now clarify the open and closed conformations as a schematic in a main text figure (new Fig. 4c). In the figure legend, we reference the original publication showing a side-by-side comparison of the actual structures (Kögel et al., 2022).

2. The exosome alone is a complicated machine with many subunits. In general, the authors did a great job of not introducing too many protein names but noted below are a few examples that could use some additional details in the text to help readers follow.

a. Line 151 – Since several subunits are highlighted in detail it would be helpful to list in () the names of the 6 PH-like proteins and the three S1/KH cap subunits.

Per the Reviewer's suggestion, we included the protein names for the RNase PH-like proteins and the S1/KH in the text.

b. Line 170 – Define that a PIN domain is an endonuclease domain and RNase II is an exonuclease domain.

We now clarified that the DIS3L subunit contains an endonuclease-like PIN domain and an RNase II-like exoribonuclease domain.

c. Line 184 (and several other places) “Gatekeeping Module”. I suggest referring to this as the “SKI2N38 Gatekeeping Module” throughout the manuscript to remind readers it includes portions of all three proteins.

We now refer to the “gatekeeping module” as the “SKI2_N38 gatekeeping module” consistently in the text.

3. If space allows it would be nice to show a side-by-side comparison of the PIN domains from Dis3 and Dis3L illustrating why the authors believe Dis3L is not active. Is there any biochemical evidence to support this? Could the inactive conformation be predicted from the alphafold model alone?

The absence of residues deemed important for activity in this PIN domain was previously noted when DIS3L was discovered as a cytoplasmic exoribonuclease (Tomecki et al., 2010). A sequence alignment is included in Tomecki et al., 2010, which we refer to now in the text. While the Reviewer raises important areas of enquiry, we believe that a more detailed comparative analysis of PIN structures may perhaps be better suited for a review.

4. In Fig 3A it is unclear what TPR30A, TPR30B, and TPR29A is referring to. Please define in the figure legend.

We included the residue numbers of each TPRs shown in Fig. 3b in the corresponding figure legend. We also noticed a mistake in the labeling while making the changes and have now also corrected TPR29A to TPR29B.

5. I’m curious about the THES disease mutations. Is there any evidence to suggest these mutations weaken the association with the ribosome or other Ski subunits? It is hard to tell in the figure, but these mutations also appear to be close to the interfaces with Ski8 and Ski2

There is currently no experimental evidence that the THES disease mutations indicated may weaken these interactions. To clarify, these disease mutants are within an integral part of the domain that interacts with the monosome in the structure, but they do not contact it directly.

Furthermore, the SKI2 arch domain is the major binding determinant for a monosome, rather than the SKI2_{N38} gatekeeping module. Future biochemical characterization of these disease mutants in SKI2_{N38} gatekeeping module would likely have to be carried out in the context of collided ribosomes.

6. In the cryo-EM data processing workflows it would be helpful to add some labels on the local resolution maps to help readers understand what is what.

We included labels on local resolution maps to help the readership in understanding those better.

Reviewer Reports on the First Revision:

Referees' comments:

Referee #1 (Remarks to the Author):

The authors have addressed most of my concerns in the revised manuscript. The additional experiments on the enrichment of SKI238 constructs on disomes, while not conclusive, provide substantial support for the proposed model. Therefore, I support the publication of the manuscript.

Referee #2 (Remarks to the Author):

The authors have suitably responded to my comments in the initial review and I am happy to recommend acceptance.

Referee #3 (Remarks to the Author):

The authors have adequately addressed my previous concerns and those of the other reviewers. The addition of experiments showing a direct interaction between SKI238 and disomes are noteworthy and will help spur new interest in the field. I fully support publication of this elegant structural and biochemical investigation into mRNA decay through the exosome-ribosome supercomplex.